# HyRF: Hybrid Radiance Fields for Memory-efficient and High-quality Novel View Synthesis

**Zipeng Wang**
zwang253@cse.ust.hk

**Dan Xu**
danxu@cse.ust.hk

Department of Computer Science and Engineering
The Hong Kong University of Science and Technology

## Abstract

Recently, 3D Gaussian Splatting (3DGS) has emerged as a powerful alternative to NeRF-based approaches, enabling real-time, high-quality novel view synthesis through explicit, optimizable 3D Gaussians. However, 3DGS suffers from significant memory overhead due to its reliance on per-Gaussian parameters to model view-dependent effects and anisotropic shapes. While recent works propose compressing 3DGS with neural fields, these methods struggle to capture high-frequency spatial variations in Gaussian properties, leading to degraded reconstruction of fine details. We present Hybrid Radiance Fields (HyRF), a novel scene representation that combines the strengths of explicit Gaussians and neural fields. HyRF decomposes the scene into (1) a compact set of explicit Gaussians storing only critical high-frequency parameters and (2) grid-based neural fields that predict remaining properties. To enhance representational capacity, we introduce a decoupled neural field architecture, separately modeling geometry (scale, opacity, rotation) and view-dependent color. Additionally, we propose a hybrid rendering scheme that composites Gaussian splatting with a neural field-predicted background, addressing limitations in distant scene representation. Experiments demonstrate that HyRF achieves state-of-the-art rendering quality while reducing model size by over 20× compared to 3DGS and maintaining real-time performance. Our project page is available at https://wzpscott.github.io/hyrf/.

## 1 Introduction

Novel view synthesis is a critical area in computer vision, with applications in scene manipulation [32, 45, 47, 33], autonomous driving [34, 43], virtual fly-throughs [44, 25, 24], and 3D generation models [19, 14, 12, 36]. Neural Radiance Fields (NeRF) [26] have emerged as a leading technology, leveraging implicit scene representations through neural networks and volume rendering to generate novel views. While NeRF-based methods excel in producing high-quality renderings with compact model sizes, they are hindered by slow rendering speeds. In recent advancements, the 3D Gaussian Splatting (3DGS) [15] method has emerged as a compelling alternative to NeRF-based approaches, enabling real-time rendering of high-resolution novel views. Unlike NeRF, which relies on continuous neural networks, 3DGS employs a set of explicit, optimizable 3D Gaussians to represent scenes. This approach is able to bypass the computational overhead of volume rendering by leveraging an efficient differentiable point-based splatting process [57, 52], achieving real-time performance while enhancing rendering quality.

However, 3DGS suffers from significant memory overhead due to its parameter-intensive representation of view-dependent colors and anisotropic shapes. Each 3D Gaussian requires 59 parameters, with 48 parameters dedicated to view-dependent color representation via spherical harmonics and 7 parameters encoding anisotropic scale and rotation. This stands in stark contrast to NeRF-based

39th Conference on Neural Information Processing Systems (NeurIPS 2025).

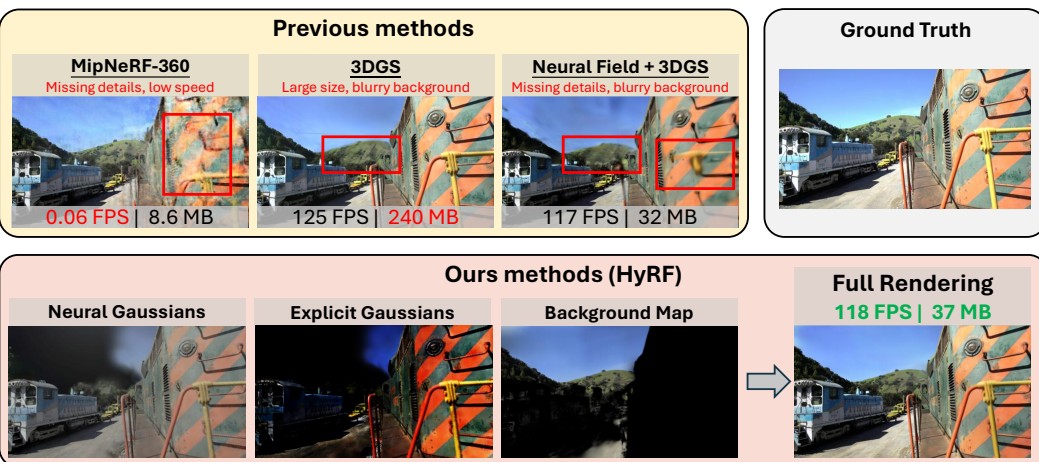

Figure 1: Mip-NeRF360 [3] struggles with inaccuracies in fine details and slow rendering speeds, while 3DGS [15] face challenges of large model sizes and blurry background. A naive combination of neural fields and 3DGS leads to loss of high-frequency information. Our method overcomes these challenges through an innovative hybrid architecture. By synergistically combining neural fields, explicit Gaussians, and neural background map, we achieve competitive or superior performance in both visual quality and model compactness, while maintaining real-time rendering capabilities.

methods, which efficiently model view-dependent effects through neural network conditioning with minimal parameter growth.

A natural approach to reducing 3DGS storage costs is to encode 3D Gaussian properties in grid-based neural fields [49, 42]. However, this method faces a fundamental limitation: the fixed resolution of grid-based representations struggles to capture the high-frequency spatial variations in 3D Gaussian properties. This issue is particularly pronounced when modeling scenes with rapid opacity and scale changes at object boundaries or high-frequency view-dependent effects. As a result, naively fitting 3D Gaussians to neural fields often fails to reconstruct fine details, such as thin geometric structures and high-frequency color variations.

In this paper, we present Hybrid Radiance Fields (HyRF), a novel scene representation that effectively addresses the frequency limitations of neural Gaussian approaches while maintaining low memory overhead. Our key insight is to decompose the representation into two complementary components: grid-based neural fields that capture low-frequency variations, and a sparse set of explicit compact Gaussians that preserve high-frequency details. Our neural component employs a decoupled architecture with two specialized neural fields: a geometry network dedicated to modeling geometric Gaussian properties (scale, opacity, and rotation), and a separate appearance network for view-dependent color prediction. This explicit disentanglement of geometric and photometric learning objectives significantly enhances representational capacity of neural fields while maintaining parameter efficiency. Meanwhile, our explicit Gaussian component stores only essential properties, i.e., 3D positions, isotropic scales, opacity values, and diffuse colors, in order to minimize memory overhead while preserving critical scene details.

To achieve both efficiency and rendering quality, we propose a hybrid rendering pipeline that operates in three stages. First, our visibility pre-culling module eliminates Gaussians outside the current view frustum, significantly reducing computational overhead of querying neural fields. Next, we process the remaining visible Gaussians by querying their positions through our neural field to predict neural Gaussian properties, which are then combined with the stored explicit parameters to recover high-frequency details. To address the insufficient background modeling of Gaussian representations, we implement a learnable solution where the neural field generates a background map projected onto a background sphere. This background map is composited with the foreground Gaussian rendering through alpha blending, therefore achieves high visual quality for both foreground and remote background objects.

In summary, our key contributions include: (i) A novel integration of neural fields with explicit compact Gaussians, preserving high-frequency details while minimizing memory overhead. (ii) A

dual-field architecture that improves the modeling of Gaussian properties by disentangling geometry and view-dependent effects. (iii) A hybrid rendering strategy that reduces computational overhead and improves rendering quality for backgrounds. (iv) Extensive experiments demonstrate that our method achieves superior rendering quality, reduces model size by 20× compared to 3DGS [15], and maintains real-time performance.

## 2   Related Work

**Neural Radiance Fields.** Neural Radiance Fields (NeRF) [26] revolutionized novel view synthesis by modeling scenes as volumetric radiance fields, where each point in space is associated with radiance and density values through a multi-layer perceptron (MLP). The state-of-the-art MLP-based method, Mip-NeRF360 [3], has achieved significant improvements in anti-aliasing and handling unbounded scenes. However, MLP-based radiance fields suffer from slow training and rendering speeds due to the extensive querying required for volume rendering. To address these inefficiencies, recent approaches have integrated NeRF with structured arrays of learnable features [21, 53, 37, 9, 41]. For instance, TensoRF [4] employs tensor decomposition to represent scenes using compact low-rank tensor components, while Instant-NGP [28] combines a multi-resolution hash table with a fully-fused MLP [27], significantly accelerating rendering. Despite these advancements, grid-based methods still face challenges in achieving real-time rendering and matching the quality of MLP-based approaches, often due to limited grid resolution or hash collisions.

**Explicit Radiance Fields.** Another line of research [1, 52, 48] explores replacing implicit neural fields with explicit, point-based scene representations, which can be rendered more efficiently using rasterization techniques. Notably, 3D Gaussian Splatting (3DGS) [15] introduced a scene representation based on 3D Gaussians, synthesizing novel views through point-based alpha blending [57]. This approach achieves state-of-the-art rendering quality and real-time performance. However, the size of models using 3D Gaussian representations is always considerably larger than NeRF-based methods.

**Compressed 3D Gaussian Splatting.** While 3D Gaussian Splatting (3DGS) achieves superior rendering performance compared to NeRF-based methods, its significantly larger model size has motivated research into compact representations that preserve its performance advantages. Existing approaches fall into two main categories: (1) parameter compression techniques using vector quantization [17, 29], and (2) hybrid neural-3DGS architectures [30, 17, 6, 42] that uses neural components to predict 3D Gaussian properties instead of explicitly storing them. Closely related to our work, Scaffold-GS [23] employs anchor points with neural features to predict local Gaussian properties, achieving superior compactness while maintaining rendering quality. Our approach differs fundamentally by predicting all Gaussian properties globally through grid-based neural fields, while augmenting high-frequency details with explicit residual Gaussians. This architecture enables both superior compression ratios and enhanced view quality. Furthermore, our method remains compatible with vector quantization techniques, achieving additional efficiency gains since our explicit Gaussians contain far fewer parameters than conventional 3DGS representations. Recently, LocoGS [40] explores a similar idea by storing Gaussian properties in neural fields. In contrast, our method stores explicit residuals for Gaussian shapes and introduces decoupled neural fields, leading to improved representation of high-frequency scene components.

## 3   Methodology

### 3.1   Preliminary: 3DGS

In 3DGS, a scene is depicted through a collection of optimizable 3D Gaussians. Each Gaussian is defined by its 3D coordinates $\mathbf{p}$, opacity $\alpha$, rotation $\mathbf{r}$, scaling factor $s$, and color $\mathbf{c}$. The opacity $\alpha$ is defined as a scalar value ranging from 0 to 1. The size of the Gaussian in 3D is indicated by scale $s$. Rotation is expressed as a quaternion $\mathbf{r}$. The color $\mathbf{c}$ uses a set of spherical harmonics to account for view-dependent effects, which is then converted into an RGB color before rasterization.

3DGS uses 3D points obtained from Structure-from-Motion libraries like COLMAP [38, 39] as initial 3D Gaussians and adaptively densifies them based on the accumulated gradients. During rendering, the 3D Gaussians are ordered by depth, projected onto 2D image planes, and combined using the

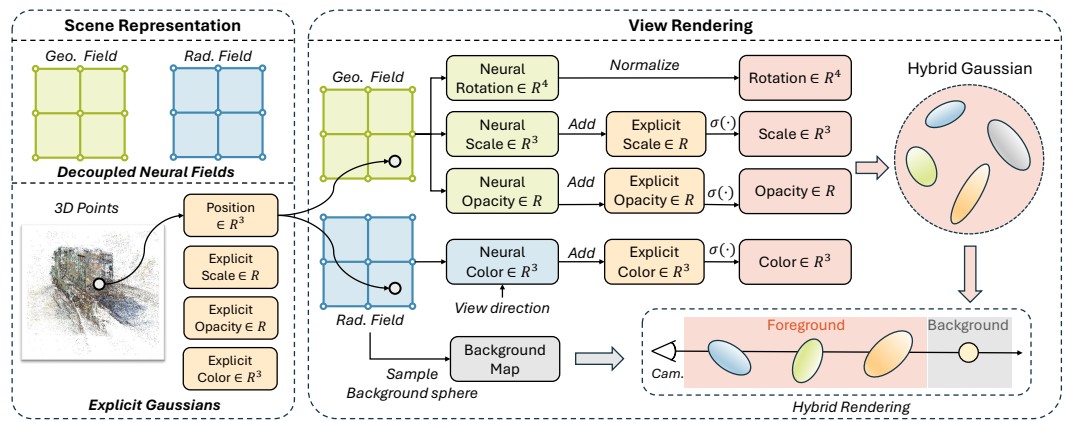

Figure 2: Framework overview. Our method represents the scene using grid-based neural fields and a set of compact explicit Gaussians storing only 3D position, 3D diffuse color, isotropic scale, and opacity. We encode the point position into a high-dimensional feature using the neural field and decode it into Gaussian properties with tiny MLP. These Gaussian properties are then aggregated with the explicit Gaussians and integrated into the 3DGS rasterizer.

following point-based alpha-blending method.

$$C = \sum_{i \in \mathcal{N}} \mathbf{c}_i \alpha_i \prod_{j=1}^{i-1} (1 - \alpha_j), \tag{1}$$

where $C$ is the final predicted pixel color, $\mathcal{N}$ is the set of sorted Gaussians projected onto the pixel.

## 3.2 Hybrid Radiance Fields

Our method represents a scene using 1) a explicit set of 3D Gaussians each holds only 8 parameters, including positions $\mathbf{p}_e \in \mathcal{R}^3$, diffuse color $\mathbf{c}_e \in \mathcal{R}^3$, isotropic scale $s_e \in \mathcal{R}$ and opacity $\alpha_e \in \mathcal{R}$. and 2) a compact grid-based neural field. We choose the multi-resolution hash encoding [28] as our neural field for its efficiency and strong performance. An overview is illustrated in Fig. 2.

**Decoupled neural fields:** Empirical results demonstrate that predicting all Gaussian properties through a single neural field fails to achieve satisfactory performance. We attribute this limitation to the weak correlation between Gaussian geometry and appearance attributes, which makes them hard to be learned jointly within a single neural field. To address this issue, we propose a decoupled neural field architecture, which predicts geometry properties (scale, opacity and rotation) and appearance property (view-dependent color) with two separate neural fields $\Theta_{\text{geo}}$ and $\Theta_{\text{rad}}$.

Given the position of a 3D point $\mathbf{p}_i$, we first employ a scene contraction technique similar to that in MipNeRF360 [3] to constrain the input coordinates. We first normalize the coordinates using the axis-aligned bounding box (AABB) $\mathbf{B}_0$ of the scene, which we defined as the minimum and maximum camera positions. Next, we contract the normalized points to the range $(0, 1)$ using the following formula:

$$\text{contract}(\mathbf{p}_i) = \begin{cases} 0.25 \cdot \mathbf{p}_i + 1 & \text{if } \|\mathbf{p}_i\| \leq 1 \\ 0.25 \cdot (2 - \frac{1}{\|\mathbf{p}_i\|})(\frac{\mathbf{p}_i}{\|\mathbf{p}_i\|}) + 1 & \text{otherwise.} \end{cases} \tag{2}$$

Note that we contract the points to $(0, 1)$ instead of $(-2, 2)$ to meet the input requirements for the multi-resolution hash [28].

Then we use the decoupled neural fields to encode it into two high-dimensional features:

$$\mathbf{f}_{\text{rad}}^i = \text{enc}(\mathbf{p}_i; \Theta_{\text{rad}}), \mathbf{f}_{\text{geo}}^i = \text{enc}(\mathbf{p}_i; \Theta_{\text{geo}}), \tag{3}$$

where $\mathbf{f}_{\text{rad}}^i$ and $\mathbf{f}_{\text{geo}}^i$ are the encoded features.

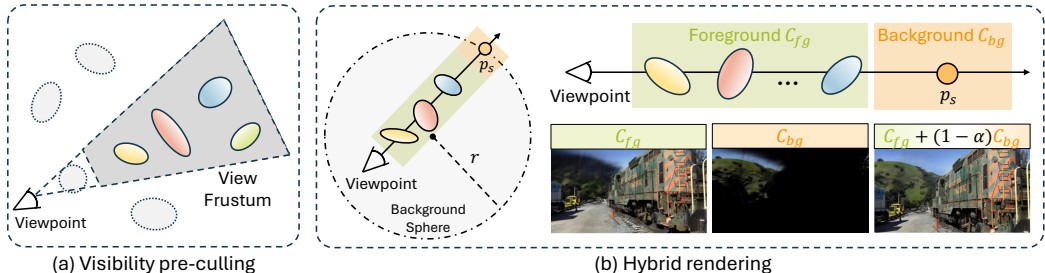

Figure 3: (a) **Visibility Pre-Culling.** We first determine whether each Gaussian lies within the current view frustum before applying neural field decoding. (b) **Hybrid Rendering Pipeline.** For each camera ray, we: (1) compute its intersection point $p_s$ with a background sphere, (2) sample the radiance field at $p_s$, and (3) composite the foreground and background colors using alpha blending.

The encoded features are then decoded into 3D Gaussian properties using two MLP-based decoders. For view-independent properties of opacity $\alpha$, scale $s$ and rotation $\mathbf{r}$, we directly decoded them as:

$$(\alpha_n, s_n, \mathbf{r}_n) = \mathrm{dec}(\mathbf{f}^i_{\mathrm{enc}}, \Phi_{\mathrm{geo}}) \tag{4}$$

To account for the view-dependent effects of Gaussian colors, we incorporate a view direction component to the MLP input using positional encoding techniques similar to NeRF-based methods [28, 3]. The view direction encoding is calculated as:

$$\mathbf{f}^i_{\mathrm{dir}} = \mathrm{PE}\left(\frac{\mathbf{p}_i - \mathbf{p}_{\mathrm{cam}}}{\|\mathbf{p}_i - \mathbf{p}_{\mathrm{cam}}\|_2}\right), \tag{5}$$

where $\mathrm{PE}(\cdot)$ is positional encoding technique [26]. The view-dependent color is decoded as:

$$\mathbf{c}_n = \mathrm{dec}(\mathbf{f}^i_{\mathrm{enc}} \oplus \mathbf{f}^i_{\mathrm{dir}}, \Phi_{\mathrm{c}}), \tag{6}$$

where $\oplus$ denotes tensor concatenation. Note the derived neural Gaussian properties $(\alpha_n, \mathbf{r}_n, s_n, \mathbf{c}_n)$ here are raw outputs from MLP without activations.

**Aggregation with explicit Gaussians:** Grid-based neural fields often overlook high-frequency scene components such as intrinsic structures. We address this problem by aggregating the predicted properties from neural fields with explicit properties stored in each Gaussian. Similar to 3DGS, we apply the sigmoid function to activate opacity and color, and use a normalization function for rotation:

$$\begin{aligned}
\alpha &= \sigma(\alpha_n + \alpha_e), \\
\mathbf{c} &= \sigma(\mathbf{c}_n + \mathbf{c}_e), \\
\mathbf{r} &= \mathrm{Normalize}(\mathbf{r}_n), \\
s &= \sigma(s_n + s_e)
\end{aligned} \tag{7}$$

where $\sigma$ denotes the sigmoid function, and $\mathrm{Normalize}(\cdot)$ denotes $L_2$ normalization of the quaternion. The aggregated Gaussian properties $(\alpha, \mathbf{r}, s, \mathbf{c})$ are then fed to the 3DGS rasterizer.

### 3.3 Hybrid Rendering

**Visibility pre-culling:** To reduce the computational overhead of querying the neural fields, we eliminate points that will not be projected onto the image plane **before** deriving their properties using the neural fields. An illustration of the visibility pre-culling process is provided in Fig. 3(a). Specifically, given a point $\mathbf{p}_i$ and a camera viewpoint, we calculate the camera-space coordinates of the point $\mathbf{p}_i$ using the camera's rotation matrix $\mathbf{R} \in \mathcal{R}^{3\times3}$ and translation vector $\mathbf{t} \in \mathcal{R}^3$ as follows:

$$\mathbf{p}_i = \mathbf{R}\mathbf{p}_i + \mathbf{t}. \tag{8}$$

We retain a point only if it is projected within the image frame, determined by the condition:

$$(|x_i| \leq 1 + \mathrm{tol}) \wedge (|y_i| \leq 1 + \mathrm{tol}), \tag{9}$$

where $x_i$ and $y_i$ are the first and second elements of $\mathbf{p}_i$, respectively. We incorporate a tolerance band $\mathrm{tol}$ in the culling process to preserve Gaussians that are partially projected outside but still intersect

with the image plane. Additionally, we discard Gaussians positioned too close to the image plane, as they may introduce optimization instability.

**Background rendering:** 3DGS often struggle to effectively densify and optimize extremely distant objects, frequently resulting in blurry backgrounds. To address this issue, we propose a hybrid rendering technique that leverages the radiance field $\Theta_{\text{rad}}$ to predict the background color. An illustration of the background rendering process is provided in Fig. 3(b).

Unlike [18], which predicts the background as points at infinity, we construct a background sphere with large radius $r$. For each ray projected from a given camera viewpoint, we compute the intersection point $\mathbf{p}_s$ between the ray and the sphere. We then use the radiance field and decoder to predict the color at point $\mathbf{p}_s$. The background color $C_{\text{bg}}$ combines the background point color with remaining visibility after accumulating the foreground Gaussians:

$$C_{\text{bg}} = \prod_{i=1}^{\mathcal{N}}(1 - \alpha_i)\mathbf{c}_s. \tag{10}$$

Finally, the pixel color is obtained by combining the foreground and background colors:

$$C = C_{\text{fg}} + C_{\text{bg}}, \tag{11}$$

where $C_{\text{fg}}$ is given by Eq. 1. In the rendering stage, we predict $C_{\text{bg}}$ only for pixels with an accumulated transmittance $T = \prod_{i=1}^{\mathcal{N}}(1 - \alpha_i)$ lower than a threshold $\tau_T$, thereby increasing rendering speed.

### 3.4 Optimization

Our method is optimized using the same L1 loss and SSIM loss [46] as the original 3DGS:

$$\mathcal{L} = (1 - \lambda)\mathcal{L}_1 + \lambda\mathcal{L}_{\text{ssim}}, \tag{12}$$

where $\lambda$ is the weight for SSIM loss. Similar the original 3DGS, we periodically reset the explicit opacity to a small value during densification and prune Gaussians with low opacity.

## 4 Experiments

### 4.1 Experimental Setup

**Dataset:** We conduct experiments on three standard real-world datasets: MipNeRF360 [3], Tanks & Temples [16], and Deep Blending [13], which together encompass a total of 13 scenes. Additionally, we utilize the NeRF Synthetic dataset [26], featuring 8 object-centered scenes. Furthermore, we examine two large-scale urban datasets captured by drones: Mill19 [44] and Urbanscene3D [20], which collectively include 4 scenes. In total, our experiments span 25 scenes across various datasets.

**Baselines:** For the MipNeRF360 [3], Tanks & Temples [16], and Deep Blending [13] datasets, we compare our method with the MLP-based NeRF method MipNeRF360 [3], two popular grid-based NeRF methods—Plenoxels [8] and Instant-NGP [28]—as well as the original 3DGS [15] and its advanced derivative, Scaffold-GS [23]. For the NeRF Synthetic dataset [26], we compare our method with MipNeRF [2], Instant-NGP [28], 3DGS [15], and Scaffold-GS [23]. For the urban-scale datasets [44, 20], we evaluate our method with two prominent NeRF-based techniques: MegaN-eRF [44] and SwitchNeRF [25], in addition to 3DGS [15] and Scaffold-GS [23]. To demonstrate the compactness of our method, we also compare a compressed version of our approach with five recent 3DGS compression methods [22, 17, 29, 6, 11, 49].

**Implementation:** Our method is built on top of the original 3DGS implementation. For the neural fields, we adopt multi-resolution hash encodings [28] with 16 levels, where each hash entry stores a feature of size 2. The maximum hash size per level for the radiance field is set to $2^{17}$ for synthetic scenes, $2^{18}$ for standard scenes, and $2^{21}$ for large scenes. The hash size for the geometry field is half that of the radiance field. For the decoder, we use a fully-fused MLP [27] with 2 hidden layers, each containing 64 neurons. For background rendering, we set the transmittance threshold $\tau_T$ to 0.2 and $r = 100$ for all scenes. All other hyperparameters remain consistent with the original 3DGS. All experiments are conducted on one NVIDIA 3090 GPU.

**Evaluation metrics:** We evaluate rendering quality of novel view synthesis using PSNR, SSIM [46], and LPIPS [56]. We also report the rendering frame rate (FPS) and model size in MB.

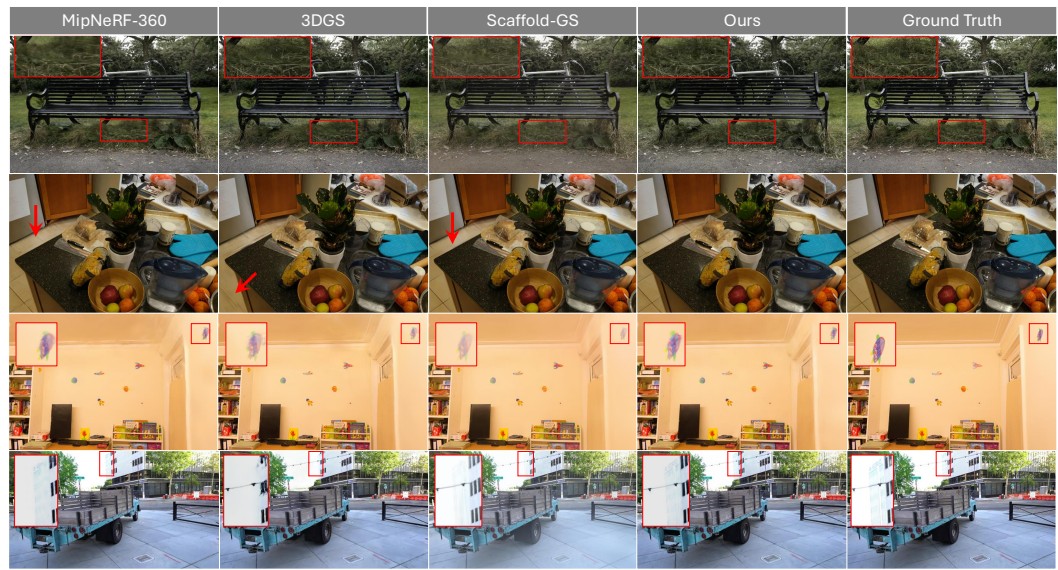

Figure 4: Qualitative comparisons of our method against previous approaches on standard real-world datasets [3, 16, 13]. The selected scenes include the *bicycle* and *counter* scenes from the MipNeRF360 dataset [2], the *playroom* scene from the DeepBlending dataset [13], and the *truck* scene from the Tanks & Temples dataset [16]. Arrows and insets are used to highlight key differences.

Table 1: Quantitative evaluation of our method compared to previous works on the MipNeRF360 [3], Tanks & Temples [16], and Deep Blending [13] datasets. We consistently achieve the *best* rendering quality, with model sizes comparable to NeRF-based methods and rendering speeds similar to 3DGS-based methods. The best results are indicated in **bold**, while the second-best results are underlined.

| Dataset | Mip-NeRF360 [3] | | | | | Tanks&Temples [16] | | | | | Deep Blending [13] | | | | |
|---|---|---|---|---|---|---|---|---|---|---|---|---|---|---|---|
| | PSNR$^\uparrow$ | SSIM$^\uparrow$ | LPIPS$^\downarrow$ | FPS$^\uparrow$ | Size(MB)$^\downarrow$ | PSNR$^\uparrow$ | SSIM$^\uparrow$ | LPIPS$^\downarrow$ | FPS$^\uparrow$ | Size(MB)$^\downarrow$ | PSNR$^\uparrow$ | SSIM$^\uparrow$ | LPIPS$^\downarrow$ | FPS$^\uparrow$ | Size(MB)$^\downarrow$ |
| Plenoxels [8] | 23.08 | 0.626 | 0.463 | 6.79 | 2150 | 21.08 | 0.719 | 0.379 | 13.0 | 2355 | 23.06 | 0.795 | 0.510 | 11.2 | 2764 |
| Instant-NGP [28] | 25.59 | 0.699 | 0.331 | 9.43 | 48 | 21.92 | 0.745 | 0.305 | 14.4 | 48 | 24.96 | 0.817 | 0.390 | 2.79 | 48 |
| M-NeRF360 [3] | 27.69 | 0.792 | 0.237 | 0.06 | **8.6** | 22.22 | 0.759 | 0.257 | 0.14 | **8.6** | 29.40 | 0.901 | 0.245 | 0.09 | **8.6** |
| 3DGS [15] | 27.21 | 0.815 | 0.214 | **117** | 734 | 23.14 | 0.841 | 0.183 | **130** | 411 | 29.41 | 0.903 | 0.243 | 112 | 676 |
| Scaffold-GS [23] | 27.39 | 0.806 | 0.252 | 86 | 244 | 23.96 | **0.853** | 0.177 | 94 | 86.5 | 30.21 | 0.906 | 0.254 | **120** | 66 |
| Ours | **27.78** | **0.816** | **0.211** | 102 | 49 | **24.02** | 0.844 | **0.176** | 106 | 39 | **30.37** | **0.910** | **0.241** | 114 | 34 |

## 4.2 Results and Evaluation

**Standard real-world scenes:** Table 1 presents the quantitative results evaluated on real-world scenes. Our method achieves state-of-the-art rendering quality while maintaining a compact model size and real-time rendering speed. Compared to 3DGS [15], our method delivers superior rendering quality while reducing the model size by over 12 times and maintaining comparable rendering speed. When compared to Scaffold-GS [23], our method shows significant improvements in rendering quality, with model sizes 1.5 to 5 times smaller and faster rendering speeds.

Qualitative comparisons between our method and previous approaches are illustrated in Fig. 4. Our method excels in capturing fine details, as demonstrated in the *bicycle*, *counter*, and *playroom* scenes, while also achieving better background modeling, as seen in the *truck* scenes.

**Object-centered synthetic scenes:** Table. 2 presents the qualitative results on the NeRF Synthetic [26] dataset. Our method achieves the best results among all the comparison methods, with a size slightly larger than Instant-NGP [28] and over 4 times smaller than 3DGS.

Table 2: Comparison on the NeRF Synthetic dataset [26].

| | PSNR$^\uparrow$ | Size(MB)$^\downarrow$ |
|---|---|---|
| MipNeRF [2] | 32.63 | **2.4** |
| Instant-NGP [50] | 33.18 | 12 |
| 3DGS [15] | 33.32 | 53 |
| Scaffold-GS [23] | 33.68 | 23 |
| Ours | **33.72** | 13 |

**Large-scale real-world scenes:** Table. 3 presents the qualitative results for two urban-scale datasets [44, 20]. Our approach achieves superior rendering quality with a more compact model size

Table 3: Quantitative evaluation of our method compared to previous works on two urban-scale datasets: Mill19 [44] and Urbanscene3D [20] dataset. Our method achieves the *best rendering quality* among all compared methods, being **4** to **7** times smaller than 3DGS-based methods and over **7000** times faster than NeRF-based methods.

| Dataset | Mill19 [44] | | | | | Urbanscene3D [20] | | | | |
|---|---|---|---|---|---|---|---|---|---|---|
| | PSNR$^{\uparrow}$ | SSIM$^{\uparrow}$ | LPIPS$^{\downarrow}$ | FPS$^{\uparrow}$ | Size(MB)$^{\downarrow}$ | PSNR$^{\uparrow}$ | SSIM$^{\uparrow}$ | LPIPS$^{\downarrow}$ | FPS$^{\uparrow}$ | Size(MB)$^{\downarrow}$ |
| MegaNeRF [44] | 22.50 | 0.55 | 0.510 | <0.01 | 32 | 23.84 | 0.699 | 0.440 | <0.01 | 32 |
| SwitchNeRF [25] | 22.93 | 0.571 | 0.485 | <0.01 | **17** | 24.54 | 0.725 | 0.418 | <0.01 | **17** |
| 3DGS [15] | 22.41 | 0.695 | 0.348 | **81** | 1566 | 21.41 | 0.763 | 0.287 | **84** | 935 |
| Scaffold-GS [23] | 22.33 | 0.658 | 0.339 | 36 | 560 | 20.25 | 0.729 | 0.295 | 34 | 435 |
| Ours | **23.52** | **0.709** | **0.319** | 75 | 215 | **24.68** | **0.791** | **0.272** | 77 | 202 |

Table 4: Quantitative evaluation of our method compared to previous 3DGS compression work on the MipNeRF-360 dataset [3].

| | PSNR$^{\uparrow}$ | SSIM$^{\uparrow}$ | LPIPS$^{\downarrow}$ | Size(MB)$^{\downarrow}$ |
|---|---|---|---|---|
| Niedermayr et al. [31] | 26.98 | 0.801 | 0.238 | 28.84 |
| Lee et al. [17] | 27.08 | 0.798 | 0.247 | 48.80 |
| Girish et al. [11] | 27.15 | 0.808 | 0.228 | 68.10 |
| Papantonakis et al. [35] | 27.1 | 0.809 | 0.226 | 25.40 |
| Chen et al. [6] | 27.59 | 0.808 | 0.234 | 22.50 |
| Ours | **27.66** | **0.814** | **0.210** | **18.04** |

Table 5: Ablation studies of the key components of our method on the Tanks & Temples dataset [16].

| | PSNR$^{\uparrow}$ | SSIM$^{\uparrow}$ | LPIPS$^{\downarrow}$ | FPS$^{\uparrow}$ | Size(MB)$^{\downarrow}$ |
|---|---|---|---|---|---|
| Full model | **24.07** | **0.847** | **0.175** | 106 | 41 |
| w/o Decouple. | 23.78 | 0.840 | 0.187 | 101 | 37 |
| w/o Explicit | 23.45 | 0.829 | 0.196 | 121 | 27 |
| w/o Neural | 22.22 | 0.797 | 0.266 | **127** | **14** |
| w/o Background | 23.43 | 0.838 | 0.19 | 112 | 41 |
| w/o Pre-culling | 24.06 | 0.847 | 0.175 | 27 | 41 |

compared to 3DGS. Notably, the gap of rendering speed between our method and 3DGS narrows as the number of rendered points increases. In contrast, Scaffold-GS experiences a significant decline in speed as the number of Gaussians grows. A qualitative comparison is can be found in the supplementary materials, where our method demonstrates a better ability to capture fine details and handle lighting variations, where 3DGS and Scaffold-GS suffers from blurs and artifacts.

**Model compression:** Though our method does not inherently include post-processing compression techniques, it remains compatible with most existing 3DGS compression approaches [22, 17]. Our representation achieves better performance by storing significantly fewer explicit Gaussian parameters. To evaluate our method's compactness, we apply post-processing techniques similar to [17], including: (1) storing point positions as half-precision tensors, (2) applying residual vector quantization (R-VQ) and Huffman encoding to explicit Gaussian properties, and (3) employing Huffman encoding with 8-bit min-max quantization for the hash table (see supplementary materials for details).

As shown in Table 4, our compressed results outperform five state-of-the-art 3DGS compression methods in both model size and rendering quality. Notably, while conventional 3DGS compression methods typically sacrifice rendering quality for storage efficiency, our approach maintains superior visual fidelity even after aggressive compression.

## 4.3 Model Analysis

**Decoupled neural fields:** We conduct a comparative analysis between our decoupled neural fields approach and a single neural field that predicts all Gaussian parameters simultaneously. To maintain experimental fairness, we configure the maximum hash size of the single neural field to be $2^{18}$, which leads to a slightly larger parameter count as our decoupled architecture. As demonstrated in Table 5, the single neural field exhibits consistent degradation across all image quality metrics. This limitation arises from the inherent challenge of using a single network to concurrently represent both geometric and appearance properties of 3D Gaussians, resulting in compromised rendering fidelity and inaccurate geometry such as gaps and holes, as visually confirmed in Fig. 5.

**Hybrid rendering:** We evaluate our model using two rendering approaches: (1) our proposed hybrid rendering pipeline and (2) conventional 3DGS rasterization. Quantitative results in Table 5 show that disabling background rendering results in significantly degraded visual quality, despite offering only marginal improvements in rendering speed. This finding supports our hypothesis that standard 3DGS approaches struggle to properly densify and optimize distant objects. As shown in Fig. 6, our qualitative analysis further reveals that background rendering plays a crucial role in maintaining high-frequency details for distant scene elements, with particularly notable of fine cloud structures.

**Neural Gaussians:** Our method leverages neural fields to predict the anisotropic shape and view-dependent color of 3D Gaussians. Without these neural components, our framework falls back to

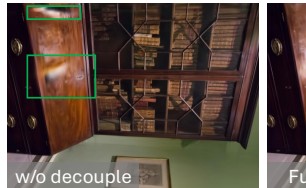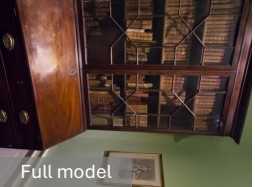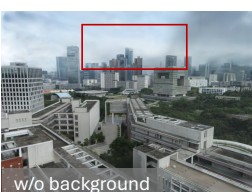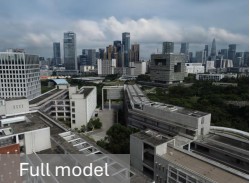

Figure 5: **Ablation of decoupled neural fields.** Using a single neural field to predict Gaussian properties causes gaps and holes.

Figure 6: **Ablation of background rendering.** The learnable background map improves the quality of distant objects (see the clouds and sky).

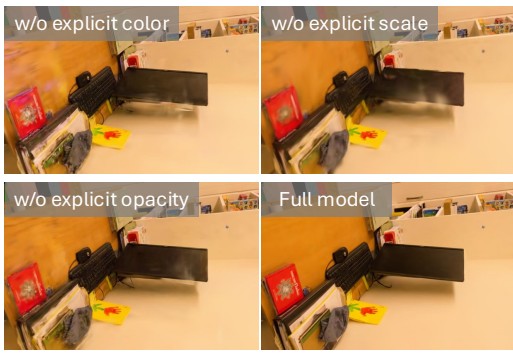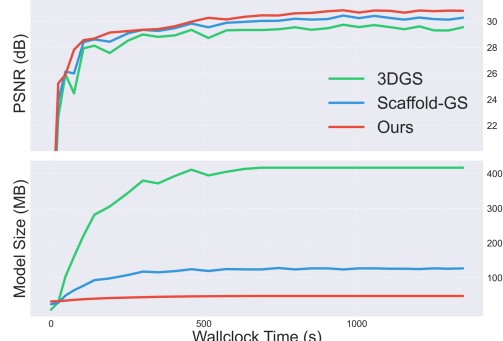

Figure 7: Detailed ablation studies of each of the explicit Gaussian properties.

Figure 8: Comparison of PSNR and model size changes during the training phase.

isotropic Gaussians with diffuse shading which has limited representation capacity, leading to a noticeable degradation in novel view synthesis quality, as demonstrated in Tab. 5.

**Visibility pre-culling:** As demonstrated in Table 5, our frustum pre-culling strategy achieves a 3.9× rendering speed improvement while maintaining equivalent visual quality for real-world 360° scenes, which represent our primary target scenario.

**Training time:** We analyze the training time of our method and compare it with other baseline methods in Fig. 8. Our method achieves significantly faster convergence, while maintaining a substantially smaller model size compared to baselines.

**Explicit Gaussians:** In Table 5, we evaluate the impact of removing all explicit Gaussian properties except positions, which are retained as they are required for neural field queries. We analyze the contribution of each explicit Gaussian component—color, scale, and opacity—through systematic ablation. Visual comparisons on the Deep Blending dataset [13] are presented in Fig. 7. Removing explicit color components causes noticeable quality deterioration, as the neural network struggles to model illumination variations and may produce unnatural colors

Table 6: Detailed ablation studies of each of the explicit Gaussian properties.

| | PSNR$^\uparrow$ | SSIM$^\uparrow$ | LPIPS$^\downarrow$ |
|---|---|---|---|
| Full model | **30.37** | **0.910** | **0.241** |
| w/o color. | 29.18 | 0.896 | 0.251 |
| w/o scale | 28.74 | 0.865 | 0.282 |
| w/o opacity | 30.21 | 0.902 | 0.247 |

due to hash collisions. The absence of explicit scale significantly impairs reconstruction of thin structures like edges and corners. We also observes removing of explicit scale often leads to instability in training. Finally, removing explicit opacity results in floaters, which also degrades output quality.

## 5   Conclusion

We have presented Hybrid Radiance Fields (HyRF), a novel approach that bridges the gap between the rendering efficiency of 3D Gaussian Splatting and the compact representation of neural fields. Our work addresses the fundamental limitations of current novel view synthesis methods by introducing a hybrid explicit-implicit representation that preserves high-frequency details, a decoupled neural field architecture that separately optimizes geometric and appearance properties, and a hybrid rendering pipeline that effectively combines the strengths of both representations. Our approach resolves the memory bottleneck of explicit Gaussian representations without sacrificing their rendering quality or

speed advantages. As novel view synthesis continues to play a crucial role in diverse applications from virtual production to autonomous systems, we believe our contributions represent a significant step toward practical, high-quality real-time neural rendering.

**Limitations:** As in the original 3DGS, our present method does not address the aliasing issue [54] and sometimes produces inaccurate surface reconstruction. Moreover, the neural field components in HyRF currently benefit from high-end GPUs for high rendering speed. Achieving comparable efficiency on web platforms or integrated graphics remains an open challenge for the community.

## Acknowledgments and Disclosure of Funding

The work is supported in part by the Early Career Scheme of the Research Grants Council (RGC) of the Hong Kong SAR under grant No. 26202321, ITF PRP/046/24FX , SAIL Research Project, and HKUST-Zeekr Collaborative Research Fund.

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

# A   Technical Appendices and Supplementary Material

## A.1   Scene Contraction

We employ a scene contraction technique similar to that in MipNeRF360 [3] to constrain the input coordinates of the multi-resolution hash to the range $(0, 1)$. First, we normalize the coordinates using the axis-aligned bounding box (AABB) $\mathbf{B}_0$ of the scene. For NeRF synthetic dataset, we set the minimum and maximum and the AABB to be -1.3 and 1.3. For standard dataset, we define the AABB using the minimum and maximum camera positions. For large-scale datasets, we use the points between the 1st and 99th percentiles of the initial point clouds to establish the AABB. The normalized point $\mathbf{p}'$ is derived as follows:

$$\mathbf{p}' = \frac{\mathbf{p}}{\mathbf{B}_0}. \tag{13}$$

Next, we contract the normalized points to the range $(0, 1)$ using the following formula:

$$\text{contact}(\mathbf{p}') = \begin{cases} 0.25 \cdot \mathbf{p}' + 1 & \text{if } \|\mathbf{p}'\| \leq 1 \\ 0.25 \cdot (2 - \frac{1}{\|\mathbf{p}'\|})(\frac{\mathbf{p}'}{\|\mathbf{p}'\|}) + 1 & \text{otherwise,} \end{cases} \tag{14}$$

where $\text{contact}()$ is the scene contraction function. Note that we contract the points to $(0, 1)$ instead of $(-2, 2)$ to meet the input requirements for the multi-resolution hash [28].

## A.2   Derivation of Ray-Sphere Intersection

In this section, we provide the detailed derivation of the ray-sphere intersection, which is used in the hybrid rendering module to compute background points. Given a ray $\mathbf{r}(t) = \mathbf{o} + t\mathbf{d}$ and a sphere centered at the origin with radius $r$, we substitute the ray equation into the sphere equation:

$$(\mathbf{o} + t\mathbf{d}) \cdot (\mathbf{o} + t\mathbf{d}) = r^2, \tag{15}$$

which expands to:

$$\mathbf{o} \cdot \mathbf{o} + 2t(\mathbf{o} \cdot \mathbf{d}) + t^2(\mathbf{d} \cdot \mathbf{d}) = r^2. \tag{16}$$

Let $A = \mathbf{d} \cdot \mathbf{d}$, $B = 2(\mathbf{o} \cdot \mathbf{d})$, and $C = \mathbf{o} \cdot \mathbf{o} - r^2$. The equation then simplifies to a quadratic in $t$:

$$At^2 + Bt + C = 0. \tag{17}$$

The solutions to this quadratic equation are given by:

$$t = \frac{-B \pm \sqrt{B^2 - 4AC}}{2A}. \tag{18}$$

Since the ray originates inside the sphere, the equation always yields two real solutions. We select the positive solution, as it corresponds to the intersection point in the forward direction of the ray.

## A.3   Ablation for View-dependent Appearance Modeling

We provide an additional ablation study that compares two approaches (SH Coefficients for "high rank per Gaussian spherical harmonics parameters" and Hybrid for "MLP and integration of neural field and explicit Gaussian") for view-dependent appearance modeling, as shown in Table. 7. Our hybrid approach not only achieves significant reduction in model size, but also achieves slightly better visual quality compared with using SH coefficients. This comparison demonstrates that our hybrid approach provides a compact and more powerful way in modeling view-dependent appearance.

Table 7: Ablation study of SH and MLP based appearance modeling.

|  | PNSR | SSIM | LPIPS | Size (MB) |
|---|---|---|---|---|
| SH Coefficients | 30.12 | 0.908 | 0.243 | 267 |
| Hybrid (Ours) | **30.37** | **0.910** | **0.241** | **34** |

## A.4 Evaluation in Street Scenes

To evaluate HyRF's performance in street scenes, we conducted experiments on the KITTI [10] dataset (2011_09_26_drive_0002 sequence), as shown in Table. 10. Our method achieves similar visual quality compared with 3DGS while being over 10 times smaller in model size. After adding the background rendering technique, our complete method shows consistent quality improvements, particularly for distant objects and sky regions.

Table 8: Evaluation in street scenes on the KITTI [10] dataset .

|  | PNSR | SSIM | LPIPS | Size (MB) |
|---|---|---|---|---|
| 3DGS | 19.37 | 0.665 | **0.272** | 472 |
| HyRF (w/o background) | 19.42 | 0.660 | 0.273 | 36.7 |
| HyRF (Full) | **19.56** | **0.667** | 0.273 | **36.4** |

## A.5 Number of Explicit Gaussians

The significant memory savings of HyRF come from both decreased per-Gaussian storage and reduced number of Gaussians. As stated in the paper, HyRF only stores 8 parameters per-Gaussian, in contrast to 59 parameters as in 3DGS. Moreover, HyRF naturally converges to fewer Gaussians while maintaining quality. As shown in Table. 11, HyRF achieves a 24-45% reduction in the number of explicit Gaussians compared to 3DGS on three dataset (MipNeRF360, Tanks&Temples and DeepBlending), without additional pruning techniques. We hypothesize this reduction of number of Gaussians stems from two key factors: (1) Faster convergence during training, reducing the need for aggressive densification, and (2) The neural field's ability to represent view-dependent effects without requiring excessive Gaussians.

Table 9: Comparison of number of explicit Gaussians.

|  | MipNeRF360 | Tanks&Temples | DeepBlending |
|---|---|---|---|
| 3DGS | 3.31M | 1.84M | 2.81M |
| HyRF | 2.52M | 1.01M | 1.74M |

## A.6 Additional Comparison with Recent 3DGS-based Methods

We conduct additional comparison experiments with several recent 3DGS-based methods, namely GOF [55], Spec-GS [51], Mini-Splatting2 [7] and DashGaussian [5] on the DeepBlending [13] dataset. To provide a more comprehensive evaluation, we have expanded the comparison table to include rendering speed (FPS), training time (Time), peak GPU memory usage (Memory), and model storage size (Size) across state-of-the-art methods.

## A.7 Additional Comparison on Specular Scenes

we have conducted additional quantitative comparisons using the anisotropic synthetic dataset from Spec-GS [51], which features 8 object-centered scenes with strong specular highlights. Compared with 3DGS, HyRF achieves significantly better rendering quality (↑1.58 dB PSNR) while using 82% less memory. The improved performance highlights the benefits of using MLPs over SH coefficients for modeling high-frequency view-dependent effects.

Table 10: Comparison with recent 3DGS-based methods.

|  | PSNR | SSIM | LPIPS | FPS | Time (min) | Memory (GB) | Size (MB) |
|---|---|---|---|---|---|---|---|
| 3DGS | 29.41 | 0.903 | 0.243 | 112 | 14.4 | 5.54 | 676 |
| GOF | 30.42 | 0.914 | 0.237 | 96 | 20.3 | 6.62 | 721 |
| Spec-GS | 30.57 | 0.912 | 0.234 | 107 | 17.8 | 5.79 | 765 |
| MiniSplatting2 | 30.08 | 0.912 | 0.240 | 136 | 2.75 | 3.65 | 155 |
| DashGaussian | 30.02 | 0.907 | 0.248 | 132 | 2.62 | 4.32 | 465 |
| HyRF | 30.37 | 0.910 | 0.241 | 114 | 12.5 | 1.83 | 34 |

Table 11: Comparison on the Spec-GS dataset.

|             | PSNR  | SSIM  | LPIPS | Size (MB) |
|-------------|-------|-------|-------|-----------|
| 3DGS        | 33.83 | 0.966 | 0.062 | 47        |
| HyRF (Ours) | 35.41 | 0.970 | 0.053 | 8.2       |

## A.8 Additional Qualitative Comparisons

In Fig. 9, we show the Additional qualitative comparisons of our method against previous approaches on standard real-world datasets.

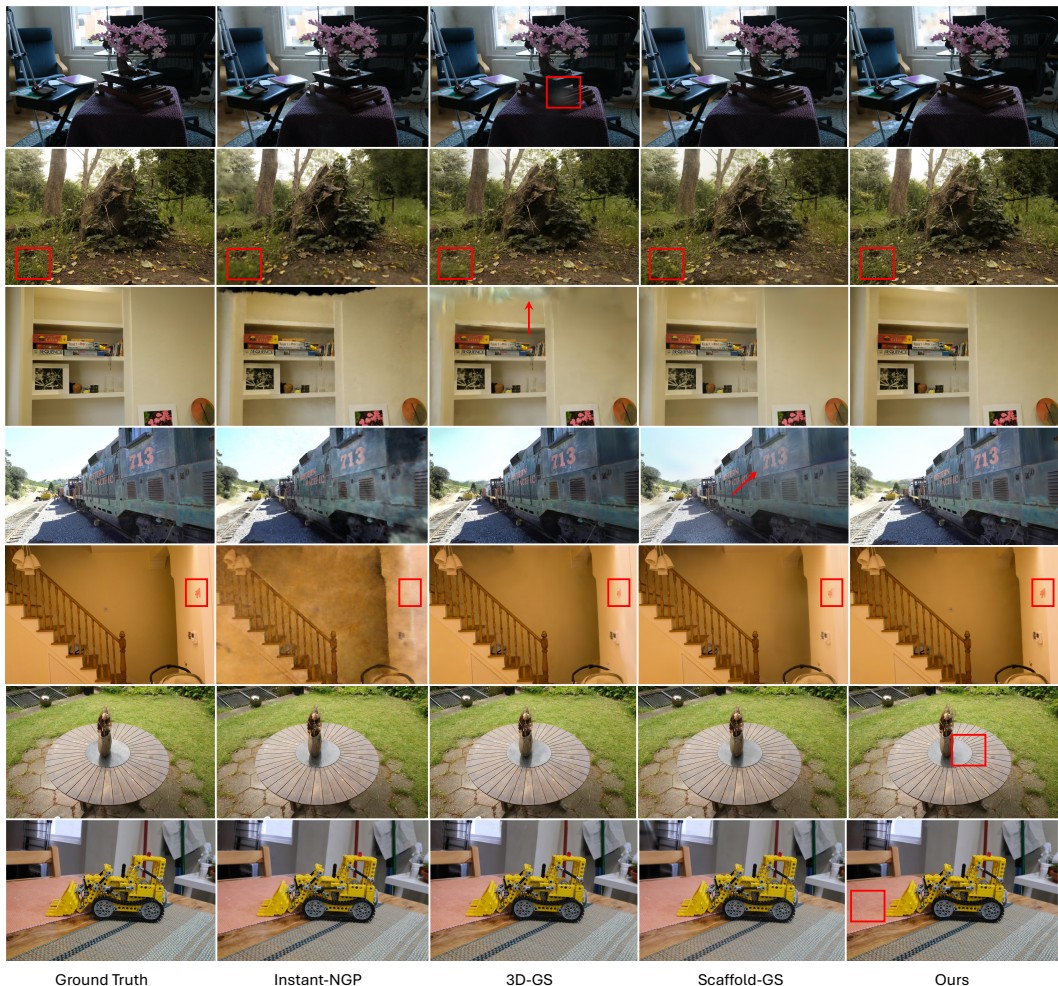

Figure 9: Additional qualitative comparisons of our method against previous approaches on standard real-world datasets.

## A.9 Per-scene Metrics

Table. 12-15 present per-scene metrics for MipNeRF360 [3], Tanks & Temples [16] and Deep Blending [13] datasets. Table. 16 and 17 provide per-scene metrics for the per-scene metrics for NeRF Synthetic dataset [26]. Finally, Table. 18 lists per-scene metrics for Mill19 [44] and Urbanscene3D [20] datasets.

Table 12: PSNR scores for scenes in Mip-NeRF360 [3], Tanks & Temples [16] and Deep Blending [13] datasets.

| dataset | | | | | Mip-NeRF360 [3] | | | | | | Tanks&Temples [16] | | | Deep Blending [13] | | |
| scene | bicycle | flowers | garden | stump | treehill | room | counter | kitchen | bonsai | mean | truck | train | mean | drjohnson | playroom | mean |
| --- | --- | --- | --- | --- | --- | --- | --- | --- | --- | --- | --- | --- | --- | --- | --- | --- |
| Plenoxels [8] | 21.91 | 20.09 | 23.49 | 20.66 | 22.24 | 27.59 | 23.62 | 23.42 | 24.66 | 23.08 | 23.22 | 18.92 | 21.08 | 23.14 | 22.98 | 23.06 |
| Instant-NGP [28] | 22.17 | 20.65 | 25.06 | 23.46 | 22.37 | 29.69 | 26.69 | 29.47 | 30.68 | 25.59 | 23.38 | 20.45 | 21.92 | 28.25 | 21.66 | 24.96 |
| M-NeRF360 [3] | 24.37 | 21.73 | 26.98 | 26.40 | 22.87 | 31.63 | **29.55** | **32.23** | **33.46** | 27.69 | 24.91 | 19.52 | 22.22 | 29.14 | 29.65 | 29.40 |
| 3DGS [15] | 25.25 | 21.52 | 27.41 | **26.55** | 22.49 | 30.63 | 28.70 | 30.32 | 31.98 | 27.21 | 25.19 | 21.10 | 23.14 | 28.77 | 30.04 | 29.41 |
| Scaffold-GS [23] | 24.50 | 21.38 | 27.17 | 26.27 | 22.44 | 31.93 | 29.34 | 31.30 | 32.70 | 27.39 | 25.77 | **22.15** | 23.96 | **29.80** | 30.62 | 30.21 |
| Ours | **25.45** | **21.56** | **27.54** | 26.19 | **22.89** | **31.98** | 29.4 | 32.01 | 33.04 | **27.78** | **25.92** | 22.12 | **24.02** | 29.72 | **31.02** | 30.37 |

Table 13: SSIM scores for scenes in Mip-NeRF360 [3], Tanks & Temples [16] and Deep Blending [13] datasets.

| dataset | | | | | Mip-NeRF360 [3] | | | | | | Tanks&Temples [16] | Deep Blending [13] | | | | |
| scene | bicycle | flowers | garden | stump | treehill | room | counter | kitchen | bonsai | mean | truck | train | mean | drjohnson | playroom | mean |
| --- | --- | --- | --- | --- | --- | --- | --- | --- | --- | --- | --- | --- | --- | --- | --- | --- |
| Plenoxels [8] | 0.496 | 0.431 | 0.606 | 0.523 | 0.509 | 0.842 | 0.759 | 0.648 | 0.814 | 0.626 | 0.774 | 0.663 | 0.719 | 0.787 | 0.802 | 0.795 |
| Instant-NGP [28] | 0.512 | 0.486 | 0.701 | 0.594 | 0.542 | 0.871 | 0.817 | 0.858 | 0.906 | 0.699 | 0.800 | 0.689 | 0.745 | 0.854 | 0.779 | 0.817 |
| M-NeRF360 [3] | 0.685 | 0.584 | 0.809 | 0.745 | 0.631 | 0.910 | 0.892 | 0.917 | 0.938 | 0.792 | 0.857 | 0.660 | 0.759 | 0.901 | 0.900 | 0.901 |
| 3DGS [15] | **0.771** | 0.605 | **0.868** | 0.775 | 0.638 | 0.914 | 0.905 | 0.922 | 0.938 | 0.815 | 0.879 | 0.802 | 0.841 | 0.899 | 0.906 | 0.903 |
| Scaffold-GS [23] | 0.705 | 0.607 | 0.842 | **0.784** | 0.620 | 0.925 | 0.914 | **0.928** | 0.946 | 0.806 | 0.883 | **0.822** | **0.853** | 0.901 | 0.904 | 0.906 |
| Ours | 0.762 | **0.611** | 0.854 | 0.756 | **0.640** | **0.930** | **0.915** | 0.927 | **0.950** | 0.816 | 0.883 | 0.806 | 0.844 | **0.904** | **0.916** | **0.910** |

Table 14: LPIPS scores for scenes in Mip-NeRF360 [3], Tanks & Temples [16] and Deep Blending [13] datasets.

| dataset | | | | | Mip-NeRF360 [3] | | | | | | Tanks&Temples [16] | | | Deep Blending [13] | | |
| scene | bicycle | flowers | garden | stump | treehill | room | counter | kitchen | bonsai | mean | truck | train | mean | drjohnson | playroom | mean |
| --- | --- | --- | --- | --- | --- | --- | --- | --- | --- | --- | --- | --- | --- | --- | --- | --- |
| Plenoxels [8] | 0.506 | 0.521 | 0.3864 | 0.503 | 0.540 | 0.4186 | 0.441 | 0.447 | 0.398 | 0.463 | 0.335 | 0.422 | 0.379 | 0.521 | 0.499 | 0.510 |
| Instant-NGP [28] | 0.446 | 0.441 | 0.257 | 0.421 | 0.450 | 0.261 | 0.306 | 0.195 | 0.205 | 0.331 | 0.249 | 0.360 | 0.305 | 0.352 | 0.428 | 0.390 |
| M-NeRF360 [3] | 0.301 | 0.344 | 0.170 | 0.261 | 0.339 | 0.211 | 0.204 | 0.127 | 0.176 | 0.237 | 0.159 | 0.354 | 0.257 | 0.237 | 0.252 | 0.245 |
| 3DGS [15] | **0.205** | 0.336 | **0.103** | 0.210 | 0.317 | 0.220 | 0.204 | 0.129 | 0.205 | 0.214 | 0.148 | 0.218 | 0.183 | 0.244 | 0.241 | 0.243 |
| Scaffold-GS [23] | 0.306 | 0.362 | 0.146 | 0.284 | 0.346 | 0.202 | 0.191 | 0.126 | 0.185 | 0.252 | 0.147 | **0.206** | 0.177 | 0.250 | 0.258 | 0.254 |
| Ours | 0.237 | **0.301** | 0.144 | 0.236 | 0.328 | **0.189** | **0.179** | **0.124** | **0.167** | 0.211 | **0.140** | 0.212 | 0.176 | **0.242** | **0.239** | **0.241** |

Table 15: Model size (MB) for scenes in Mip-NeRF360 [3], Tanks & Temples [16] and Deep Blending [13] datasets.

| dataset | | | | | Mip-NeRF360 [3] | | | | | | Tanks&Temples [16] | | | Deep Blending [13] | | |
| scene | bicycle | flowers | garden | stump | treehill | room | counter | kitchen | bonsai | mean | truck | train | mean | drjohnson | playroom | mean |
| --- | --- | --- | --- | --- | --- | --- | --- | --- | --- | --- | --- | --- | --- | --- | --- | --- |
| 3DGS [15] | 1291 | 1045 | 1268 | 1034 | 872 | 327 | 261 | 414 | 281 | 634 | 578 | 240 | 411 | 715 | 515 | 676 |
| Scaffold-GS [23] | 248 | 217 | 271 | 493 | 209 | 133 | 194 | 173 | 258 | 244 | 107 | 66 | 87 | 69 | 63 | 66 |
| Ours | **68** | **55** | **54** | **51** | **61** | **41** | **39** | **38** | **38** | **49** | **41** | **37** | **39** | **48** | **36** | **34** |

Table 16: PSNR scores for scenes in Synthetic NeRF dataset [26].

| scene | Mic | Chair | Ship | Materials | Lego | Drums | Ficus | Hotdog | mean |
| --- | --- | --- | --- | --- | --- | --- | --- | --- | --- |
| 3DGS [15] | 35.36 | 35.83 | 30.80 | 30.00 | 35.78 | 26.15 | 34.87 | 37.72 | 33.32 |
| Scaffold-GS [23] | **37.25** | 35.28 | 31.17 | **30.65** | 35.69 | 26.44 | 35.21 | 37.73 | 33.68 |
| Ours | 35.91 | **35.51** | 31.76 | 30.13 | **36.30** | 26.44 | 35.60 | 38.18 | 33.72 |

Table 17: Model size for scenes in Synthetic NeRF dataset [26].

| scene | Mic | Chair | Ship | Materials | Lego | Drums | Ficus | Hotdog | mean |
| --- | --- | --- | --- | --- | --- | --- | --- | --- | --- |
| 3DGS [15] | 50 | 116 | 63 | 35 | 78 | 93 | 59 | 44 | 53 |
| Scaffold-GS [23] | 12 | 13 | 16 | 18 | 13 | 35 | **11** | **8** | 23 |
| Ours | **11.2** | **10.9** | **12.2** | **13.6** | **12.1** | **13.2** | 14.7 | 12.4 | **13** |

Table 18: Per-scene metrics on Mill19 [44] dataset.

| Scene | Rubble | | | | Building | | | | Sci-art | | | | Residence | | | |
| | PSNR$^\uparrow$ | SSIM$^\uparrow$ | LPIPS$^\downarrow$ | Size$^\downarrow$ | PSNR$^\uparrow$ | SSIM$^\uparrow$ | LPIPS$^\downarrow$ | Size$^\downarrow$ | PSNR$^\uparrow$ | SSIM$^\uparrow$ | LPIPS$^\downarrow$ | Size$^\downarrow$ | PSNR$^\uparrow$ | SSIM$^\uparrow$ | LPIPS$^\downarrow$ | Size$^\downarrow$ |
| --- | --- | --- | --- | --- | --- | --- | --- | --- | --- | --- | --- | --- | --- | --- | --- | --- |
| 3DGS [15] | 24.21 | 0.695 | 0.357 | 1566 | 20.6 | 0.677 | 0.340 | 1424 | 21.84 | 0.801 | 0.279 | 596 | 20.97 | 0.726 | 0.295 | 1273 |
| Scaffold-GS [23] | 22.69 | 0.662 | 0.342 | 521 | 19.97 | 0.655 | 0.3367 | 599 | 18.9 | 0.763 | 0.286 | 303 | 19.6 | 0.695 | 0.303 | 567 |
| Ours | **25.3** | **0.709** | **0.331** | **183** | **21.75** | **0.710** | **0.305** | **194** | **26.07** | **0.830** | **0.247** | **123** | **23.28** | **0.751** | **0.295** | **162** |

