# OpenReview forum: "HyRF: Hybrid Radiance Fields for Memory-efficient and High-quality Novel View Synthesis"
_NeurIPS.cc/2025/Conference — NeurIPS 2025 poster_

### Official Review · Reviewer_C7xm · 2025-06-25

**Clarity:** 3
**Significance:** 3
**Originality:** 3
**Rating:** 5
**Confidence:** 4

**Summary:**

The paper introduces Hybrid Radiance Fields (HyRF), a novel scene representation that combines explicit 3D Gaussians with grid-based neural fields to enable real-time, high-fidelity novel view synthesis while achieving a 20× reduction in memory usage compared to traditional 3DGS. Key contributions include a decoupled neural architecture for separate geometry and appearance prediction, and a hybrid rendering strategy that fuses Gaussian splatting with neural background projection. This design effectively addresses the slow rendering of NeRF-based methods and the memory inefficiency of 3DGS. Extensive experiments are conducted to validate the superiority of HyRF, demonstrating both high-quality rendering and high efficiency.

**Questions:**

Since 3DGS falls short in modeling specular regions, it would be valuable to assess how HyRF performs in such scenarios. The current evaluation only includes quantitative comparisons on reflective cases such as the Materials scene in the NeRF Synthetic dataset. Including more qualitative comparisons on specular scenes would strengthen the evaluation and provide better demonstration of the model’s ability in handling specular scenes.

**Ethical Concerns:**

["NO or VERY MINOR ethics concerns only"]

**Final Justification:**

The authors addressed all my concerns during the rebuttal period, so I would maintain my positive rating.

**Limitations:**

The authors have adequately addressed the limitations in the paper. Limitations include the aliasing issue and inaccurate surface reconstruction.

**Paper Formatting Concerns:**

No major formatting issues in the paper.

**Quality:**

3

**Strengths And Weaknesses:**

Strengths:
1. The proposed HyRF is a novel representation that achieves high-quality rendering while minimizing memory overhead. The approach achieves a substantial model compression of 12–20× compared to 3DGS while maintaining real-time rendering performance.
2. The decoupled neural fields for geometry and appearance prediction effectively mitigates the challenges of joint learning, improving the modeling of view-dependent appearance and background. This design choice is validated by ablation studies showing a 0.29 PSNR drop when reverting to a unified field.
3. The proposed neural background projection addresses the issue of blurry distant objects in 3DGS. Both quantitative and qualitative comparisons demonstrate the effectiveness of this implementation.

Weaknesses:
1. The overall pipeline largely builds upon existing components like Instant-NGP (grid-based) and 3DGS (gaussians), with several practical implementation enhancements. While this integration somewhat limits the novelty of the method, the idea of fusing grid-based neural fields with 3DGS is conceptually novel and has been explored in a thoughtful and effective manner. In addition, the resulting performance is impressive, and the design choices are effective and well justified.
2. It is recommended that the authors include a detailed comparison of memory consumption across different methods to provide a more comprehensive evaluation.
3. To strengthen the experimental evaluation, it is recommended to include more recent 3DGS-based methods such as GOF, Spec-GS and Mini-Splatting2, DashGaussian, which offer complementary strengths in rendering quality and efficiency correspondingly.

---

> ### Author Rebuttal · Authors · 2025-07-31
>
> > The overall pipeline largely builds upon existing components like Instant-NGP (grid-based) and 3DGS (gaussians), with several practical implementation enhancements. While this integration somewhat limits the novelty of the method, the idea of fusing grid-based neural fields with 3DGS is conceptually novel and has been explored in a thoughtful and effective manner. In addition, the resulting performance is impressive, and the design choices are effective and well justified.
>
> We sincerely appreciate the reviewer's recognition of the novelty and contributions of our work.
> To provide a further illustration, we believe our core innovation lies in the novel framework that decomposes Gaussian properties between compact per-Gaussian storage and spatially-queryable neural fields, which enables new trade-offs between memory, quality, and speed that weren't previously achievable. Extensive experiments demonstrate that our method achieves superior rendering quality, reduces model size by 20 times compared to 3DGS, and maintains real-time performance.
>
> ---
>
> > It is recommended that the authors include a detailed comparison of memory consumption across different methods to provide a more comprehensive evaluation.
>
> We thank the reviewer for this valuable suggestion. We provide a detailed model storage size (MB) breakdown comparing HyRF with 3DGS on the DeepBlending dataset in the attached table. HyRF replaced the memory-consuming view-dependent colors and anisotropic shapes in 3DGS with compact neural fields, which led to a significant reduction in the model storage size.
>
> |  | Pos. | Opa. | Sca. | Rot. | Col. | Rad. Field | Geo. Field | Total |
> | :---- | :----: | :----: | :----: | :----: | :----: | :----: | :----: | :----: |
> | 3DGS | 34.4 | 11.5 | 34.4 | 45.8 | 550.4 | \- | \- | 676 |
> | HyRF | 10.5 | 3.5 | 3.5 | \- | 10.5 | 3.7 | 1.8 | 34 |
>
> ---
>
> > To strengthen the experimental evaluation, it is recommended to include more recent 3DGS-based methods such as GOF, Spec-GS and Mini-Splatting2, DashGaussian, which offer complementary strengths in rendering quality and efficiency correspondingly.
>
> We conduct comparison experiments with GOF \[1\], Spec-GS \[2\], Mini-Splatting2 \[3\] and DashGaussian \[4\] on the DeepBlending dataset, which will be included in our revised version.
>
> * Compared with GOF and Spec-GS, our method offers slightly lower visual quality (within 0.2 PSNR) with more than 20 times smaller model size. Notably, their enhancement techniques (GOF's surface modeling, Spec-GS's ASG encoding) could potentially be integrated into our framework.
> * Compared with MiniSplatting2 and DashGaussian, HyRF outperforms Mini-Splatting2 and DashGaussian in both visual quality and model compactness, demonstrating the advantages of our hybrid representation approach. Their training acceleration techniques also have potential to be integrated into our framework.
>
> |  | PSNR | SSIM | LPIPS | Size (MB) |
> | :---- | :----: | :----: | :----: | :----: |
> | GOF | 30.42 | **0.914** | 0.237 | 721 |
> | Spec-GS | **30.57** | 0.912 | **0.234** | 765 |
> | MiniSplatting2 | 30.08 | 0.912 | 0.240 | 155 |
> | DashGaussian | 30.02 | 0.907 | 0.248 | 465 |
> | HyRF | 30.37 | 0.910 | 0.241 | **34** |
>
> ---
>
> > Since 3DGS falls short in modeling specular regions, it would be valuable to assess how HyRF performs in such scenarios. The current evaluation only includes quantitative comparisons on reflective cases such as the Materials scene in the NeRF Synthetic dataset. Including more qualitative comparisons on specular scenes would strengthen the evaluation and provide better demonstration of the model’s ability in handling specular scenes.
>
> We appreciate the reviewer's valuable suggestion regarding evaluation on specular scenes. While we are not allowed to include new visual results in this rebuttal, we have conducted additional quantitative comparisons using the anisotropic synthetic dataset from \[2\], which features 8 object-centered scenes with strong specular highlights.
> Compared with 3DGS, HyRF achieves significantly better rendering quality (↑1.58 dB PSNR) while using 82% less memory. The improved performance highlights the benefits of using MLPs over SH coefficients for modeling high-frequency view-dependent effects. This quantitative comparison, together with additional qualitative comparisons on specular scenes will be added to our revised version.
>
> |  | PSNR | SSIM | LPIPS | Size (MB) |
> | :---- | :----: | :----: | :----: | :----: |
> | 3DGS | 33.83 | 0.966 | 0.062 | 47 |
> | HyRF | **35.41** | **0.970** | **0.053** | **8.2** |
>
> ---
>
> \[1\] Gaussian Opacity Fields: Efficient Adaptive Surface Reconstruction in Unbounded Scenes. SIGGRAPH ASIA 2024\.
> \[2\] Spec-Gaussian: Anisotropic View-Dependent Appearance for 3D Gaussian Splatting. NeurIPS2024.
> \[3\] Mini-Splatting2: Building 360 Scenes within Minutes via Aggressive Gaussian Densification. ArXiv.
> \[4\] DashGaussian: Optimizing 3D Gaussian Splatting in 200 Seconds. CVPR2025.

---

> > ### Comment · Reviewer_C7xm · 2025-08-02
> >
> > Thank you for your response and most of my concerns have been addressed. I would suggest expanding the table to cover GPU memory consumption, FPS, training time, and model storage size across different methods for a more comprehensive evaluation.

---

> > > ### Author Response · Authors · 2025-08-03
> > >
> > > We sincerely appreciate the reviewer’s constructive feedback. To provide a more comprehensive evaluation, we have expanded the comparison table to include rendering speed (*FPS*), training time (*Time*), peak GPU memory usage (*Memory*), and model storage size (*Size*) across state-of-the-art methods.
> > > As shown in the updated table, our method (HyRF) achieves significant reductions in memory consumption and model size due to our hybrid design, while maintaining competitive reconstruction quality and rendering speed. Although our training time is longer than methods like MiniSplatting2 and DashGaussian that specifically optimize training efficiency, we emphasize that our framework is complementary to such acceleration techniques and could readily integrate them for further improvements.
> > >
> > > |  | PSNR | SSIM | LPIPS | FPS | Time (min) | Memory (GB) | Size (MB) |
> > > | :---- | :----: | :----: | :----: | :----: | :----: | :----: | :----: |
> > > | 3DGS | 29.41 | 0.903 | 0.243 | 112 | 14.4 | 5.54 | 676 |
> > > | GOF | 30.42 | **0.914** | 0.237 | 96 | 20.3 | 6.62 | 721 |
> > > | Spec-GS | **30.57** | 0.912 | **0.234** | 107 | 17.8 | 5.79 | 765 |
> > > | MiniSplatting2 | 30.08 | 0.912 | 0.240 | **136** | 2.75 | 3.65 | 155 |
> > > | DashGaussian | 30.02 | 0.907 | 0.248 | 132 | **2.62** | 4.32 | 465 |
> > > | HyRF | 30.37 | 0.910 | 0.241 | 114 | 12.5 | **1.83** | **34** |

---

> > > > ### Comment · Reviewer_C7xm · 2025-08-03
> > > >
> > > > Thank you for your response and all my concerns have been addressed. I would suggest including this table in the final version of paper. I would maintain my positive rating.

---

> > > > > ### Author Response · Authors · 2025-08-03
> > > > >
> > > > > We sincerely appreciate the reviewer's positive feedback and constructive suggestion. We would like to confirm that the extended table will be included in the final version of the paper. We thank the reviewer for maintaining their positive assessment of our work.

---

### Official Review · Reviewer_HRRA · 2025-07-01

**Clarity:** 3
**Significance:** 4
**Originality:** 4
**Rating:** 5
**Confidence:** 4

**Summary:**

The paper introduces HyRF, a hybrid explicit–neural Gaussian-splatting scheme that stores minimal per-Gaussian properties explicitly, deferring richer encoding of properties to a neural field queried only at visible Gaussians. This design yields comparable rendering speeds to 3DGS, improves compression over Scaffold-GS, and simplifies the pipeline by avoiding anchor points or other compressed encodings of Gaussian positions.

**Questions:**

1. Why is Scaffold-GS the only compact or reduced memory Gaussian splatting method that is compared against in the majority of experiments? It may help to justify or explain this choice briefly in the “baselines” section.

2. One of the claims of the paper is that only visible Gaussians need to query the neural field, which reduces the rasterization time. Does this benefit apply at all to object centered scenes where most Gaussians are within the view frustum?

3. What limitations if any does storing the rotation entirely within the neural field impose? This seems to be distinct from all other Gaussian parameters which are entirely (e.g. position) or partially (e.g. scale, opacity, color) represented by an explicit Gaussian property.

Having a better understanding of the choice for baselines and the limitations of the method would increase my score for clarity.

**Ethical Concerns:**

["NO or VERY MINOR ethics concerns only"]

**Final Justification:**

The paper presents a clear and well-motivated hybrid explicit–neural Gaussian splatting approach that achieves strong compression, competitive rendering speed, and improved PSNR over state of the art, while simplifying the representation by removing anchors and explicit Gaussian positions. The decomposition of Gaussian properties into lightweight explicit storage and neural-field-predicted components is elegant and original. My main questions--on baseline selection, benefits in object-centered scenes, and the implications of storing rotation in the neural field--were addressed in the rebuttal with convincing clarifications. With these points resolved, I maintain my positive recommendation for acceptance.

**Limitations:**

The paper could benefit from more detailed and in-depth discussion of the limitations, which currently are limited to a single somewhat general sentence at the end of the paper. I'm generally positive about the method, but understanding the method’s shortcomings or limitations--even if minor--would be helpful for future researchers or practitioners who want to build on this method.

**Paper Formatting Concerns:**

No major formatting issues.

**Quality:**

4

**Strengths And Weaknesses:**

strengths:
1. HyRF outperforms Scaffold-GS in both model compression and render speed, while removing the need for anchor points or predicted Gaussian positions.
2. Clear, well-motivated exposition with extensive ablations.
3. Empirical results show higher PSNR than state of the art neural renderers with runtime similar to 3DGS and smaller model size than Scaffold-GS.
4. Decomposing Gaussian properties into a lightweight component stored per-Gaussian and a more expressive component queried spatially is an original and elegant approach. The closest comparison in prior works are methods like ([14] Lee et al. 2024) which use separate representations but only for entirely separate properties (e.g. explicit Gaussians for geometric properties and neural field for appearance).

weaknesses:
1. The paper only briefly states that limitations mirror those of 3DGS; more details on when HyRF breaks down or what it struggles with would be useful.

---

> ### Author Rebuttal · Authors · 2025-07-27
>
> > The paper only briefly states that limitations mirror those of 3DGS; more details on when HyRF breaks down or what it struggles with would be useful.
>
> We appreciate the reviewer's suggestion to elaborate on HyRF's limitations. Our method currently has several potential limitations that warrant discussion:
>
> 1. Like 3DGS, our current formulation doesn't directly produce accurate surfaces. Future adaptations could incorporate surfel-based representations (e.g.  2DGS \[1\]) with our work.
> 2. The rendering pipeline currently does not incorporate anti-aliasing, which could be improved through similar techniques like MipSplatting \[2\].
> 3. We employ a consistent hash size for all standard datasets, which yields satisfactory results. However, the optimal hash size may vary depending on the per-scene scale and complexity. This may slightly increase the complexity of hyper-parameter tuning compared with the original 3DGS.
>
> We will expand our limitations discussion in the revised manuscript to include these important considerations, along with their potential mitigation strategies.
>
> ---
>
> > Why is Scaffold-GS the only compact or reduced memory Gaussian splatting method that is compared against in the majority of experiments? It may help to justify or explain this choice briefly in the “baselines” section.
>
> We focus our primary comparisons (e.g. Table 1\) on 3DGS and Scaffold-GS as they represent the two mainstream paradigms in Gaussian splatting: point-based (3DGS) and anchor-based (Scaffold-GS) approaches. These two paradigms encompass most current Gaussian splatting variants, making them ideal baselines for evaluating our novel hybrid representation. *We include additional comparisons with specialized compression methods in Table 4 to demonstrate our advantages over Gaussian compression approaches as well.*
>
> We note that many compression techniques from these compression methods (e.g., quantization, pruning, or entropy coding) could potentially be combined with our method for further memory reduction. However, our current experiments focus on evaluating the core contributions of our hybrid neural field approach relative to these fundamental baselines. We will clarify this rationale in the "baselines" section as suggested.
>
> ---
>
> > One of the claims of the paper is that only visible Gaussians need to query the neural field, which reduces the rasterization time. Does this benefit apply at all to object centered scenes where most Gaussians are within the view frustum?
>
> The reviewer raises a valid point regarding object-centered scenes. Indeed, the performance benefit from visibility pre-culling is less pronounced for such cases (e.g., NeRF Synthetic dataset) where most Gaussians remain within the view frustum. However, for real-world 360° scenes, which represent our primary target scenario, visibility pre-culling provides substantial efficiency gains, as typically only 5-10% of Gaussians are visible in any given view. As evidenced by our ablation studies in Table 5, this optimization is crucial for maintaining real-time performance in real-world environments. We will clarify this distinction between object-centered and 360° scenes in our revised manuscript.
>
> ---
>
> > What limitations if any does storing the rotation entirely within the neural field impose? This seems to be distinct from all other Gaussian parameters which are entirely (e.g. position) or partially (e.g. scale, opacity, color) represented by an explicit Gaussian property.
>
> We appreciate the reviewer's insightful question about our design choice for rotation representation. Our decision to encode rotations entirely within the neural field rather than storing them explicitly was based on the following reasons:
>
> 1. **Performance Impact:** Our experiments show that explicit rotation storage provides only marginal quality improvements (see the attached table), suggesting that the neural field effectively learns to predict rotations without significant accuracy loss.
> 2. **Memory Efficiency:** Representing rotations explicitly would require storing 4 additional parameters per Gaussian (for quaternion representation), increasing memory usage by approximately 30% with minimal benefit.
> 3. **Design Consistency:** This approach maintains consistency with our overall framework where anisotropic effects like rotations are handled by the neural fields, while isotropic properties remain explicit.
>
> The attached table compares our full model (denoted as *HyRF*) and an alternative which additionally stores a per-Gaussian rotation property (denoted as *HyRF \+ Rot*.). The comparison demonstrates that the neural-field-predicted rotations achieve similar rendering quality while maintaining superior memory efficiency. We believe this represents a favorable trade-off for most practical applications.
>
> |  | PSNR | SSIM | LPIPS | Size (MB) |
> | :---- | :----: | :----: | :----: | :----: |
> | HyRF | 30.37 | 0.910 | 0.241 | 33.9 |
> | HyRF \+ Rot. | 30.38 (+0.01) | 0.910 (-0) | 0.240 (-0.001) | 45.5 (+11.5) |
>
> ---
>
> \[1\] 2D Gaussian Splatting for Geometrically Accurate Radiance Fields. SIGGRAPH 2024\.
> \[2\] Mip-Splatting: Alias-free 3D Gaussian Splatting. CVPR 2024\.

---

> > ### Comment · Reviewer_HRRA · 2025-08-05
> >
> > Thank you for the thoughtful responses; all of my questions have been addressed and I maintain my positive rating.

---

### Official Review · Reviewer_Lw1w · 2025-07-01

**Clarity:** 2
**Significance:** 3
**Originality:** 2
**Rating:** 4
**Confidence:** 4

**Summary:**

This paper introduces a hybrid scene representation that combines neural fields and explicit Gaussians to achieve memory-efficient and high-quality novel view synthesis. Specifically, each Gaussian point contains only a few explicit properties (8 dimensions): 3D position, 3D diffuse color, isotropic scale, and opacity. Neural fields are applied to query the neural properties for each point, such as neural rotation, neural anisotropic scale, neural opacity, and neural color. Finally, the explicit properties and neural properties are added and activated to obtain the final properties of the Gaussian points. This approach significantly reduces the number of parameters for each Gaussian, thereby lowering memory requirements and achieving a memory-efficient scene representation.

Although the introduction of neural fields' query operations increases computation, the method incorporates visibility pre-culling to accelerate rendering. It also leverages a common technique, background rendering, used in NeRF-based unbounded scene novel view synthesis, to enhance rendering quality in distant areas.

Experiments demonstrate that this method pushes the boundaries of rendering quality and memory efficiency while maintaining high rendering speed. Moreover, if this representation is further compressed, it achieves better rate-distortion performance compared to representative works in 3DGS compression.

**Questions:**

1. **Activation function on scale**: It is not common to use a sigmoid function for the scale value in 3DGS. Could the authors explain why they choose a sigmoid function instead of the original exponential function used in 3DGS?

2. **Report of the number of explicit Gaussians**: The reviewer believes that when discussing memory-efficient scene representation, it is important not only to report the final memory size but also to include the number of explicit Gaussians. This would allow readers to understand whether the proposed method reduces memory usage primarily by decreasing the data per Gaussian or also by reducing the number of points used for scene representation.

If the authors provide clear explanations for my questions and promise to revise areas of the paper where the writing is not polished, I will consider upgrading my evaluation from borderline reject to borderline accept.

**Ethical Concerns:**

["NO or VERY MINOR ethics concerns only"]

**Final Justification:**

I appreciate the authors' thorough and careful rebuttal, which has addressed most of my concerns. Regarding the novelty issue, since more than one reviewer has recognized this paper's contributions, I've decided not to be overly strict on this point. However, I still want to highlight a limitation: since this work involves neural fields as a key component for scene representation, it cannot achieve efficient real-time rendering on web platforms or low-end consumer GPUs. I hope the authors will clearly state this limitation in the final version.

Finally, I've decided to raise my rating from borderline reject to borderline accept.

**Limitations:**

1. **Inefficiency on web platforms or low-end consumer GPUs**: The authors have made efforts in methodology and engineering to enable HyRF to achieve FPS comparable to vanilla 3DGS on an Nvidia 3090 GPU. However, since HyRF relies on Neural Fields during rendering, and Neural Fields tend to have low execution efficiency on web platforms or low-end GPUs, it is likely that HyRF's rendering efficiency on such devices will be significantly lower than that of vanilla 3DGS. This is a potential issue but does not detract from the fact that this is a well-rounded and meaningful research paper. This point is simply raised to highlight potential challenges in practical applications.

**Paper Formatting Concerns:**

There is a typo at the end of Line 242. Additionally, the writing of formulas in the paper requires improvement. Clear and consistent notation should be ensured, and all symbols must be properly defined to enhance readability and precision.

**Quality:**

3

**Strengths And Weaknesses:**

## Strengths
1. **Well-rounded solution and strong performance**: The proposed solution is complete, covering areas from representation design to rendering acceleration, background rendering quality improvement, and even model compression. It achieves a well-balanced result in rendering quality, memory efficiency, and rendering speed. Compared to the two important baselines, 3DGS and Scaffold-GS, it also demonstrates advantages in training convergence speed and model size.

2. **Interesting representation design**: Although the reviewer considers the concept of querying attributes of Gaussian in neural fields to be somewhat trivial, this work still introduces some interesting and reasonable design ideas. For instance, the use of decoupled neural fields—geometry and appearance neural fields—to extract different attributes. Additionally, each Gaussian is initialized with an isotropic scale and then refined by the geometry neural field to describe the specific shape of the Gaussian.

## Weakness
1. **Limited novelty or technical contribution**: While the overall solution is very comprehensive, most of the techniques, aside from Decoupled Neural Fields and Aggregation with Explicit Gaussians, are relatively common or lack significant innovation. As a result, the impression on the reviewer is that this is a complete but somewhat lacking in novelty work. This might be a subjective bias of the reviewer and could be further discussed with the AC and other reviewers.
2. **Limitations in formula expression.**：The reviewer places great emphasis on the accuracy of formulas. However, this paper repeatedly contains typos or parts that are not easy to follow. For example, the "isotropic scale" on Line 120 is evidently a scalar and should not be in bold. Similarly, in Eqn (2), the symbol p′p'p′ has not appeared earlier and is not explained in the surrounding text. In Eqn (3), it is clear that the outputs fradf_{\text{rad}}frad​ and fgeof_{\text{geo}}fgeo​ should be vectors. The introduction of Eqn (10) is overly casual; although the reviewer can infer the intended meaning, such writing issues should not appear in a NeurIPS paper.

---

> ### Author Rebuttal · Authors · 2025-07-31
>
> > **Limited novelty or technical contribution**: While the overall solution is very comprehensive, most of the techniques, aside from Decoupled Neural Fields and Aggregation with Explicit Gaussians, are relatively common or lack significant innovation. As a result, the impression on the reviewer is that this is a complete but somewhat lacking in novelty work. This might be a subjective bias of the reviewer and could be further discussed with the AC and other reviewers.
>
> We appreciate the reviewer's thoughtful assessment and would like to highlight three key aspects of our contribution:
>
> 1. **Novel framework.** Our core innovation lies in the novel decomposition framework that decomposes Gaussian properties between (1) compact per-Gaussian properties and (2) spatially-queryable neural fields, which conceptually enables significant memory reduction over prior art.
> 2. **Novel techniques.** As recognized by the reviewer, we develop several technical advances, including decoupled neural fields, visibility pre-culling, and background rendering. These techniques synergistically support this framework, leading to performance improvements in terms of quality and efficiency.
> 3. **Strong performance.** Our comprehensive experiments demonstrate significant advantages over existing methods in memory efficiency while maintaining similar or superior rendering quality and rendering speed.
>
> As noted by Reviewers HRRA and C7xm, our approach establishes a new paradigm for efficient neural rendering that achieves capabilities beyond prior art. We believe our novel combination and systematic framework represents a conceptual and practical advance in the field.
>
> ---
>
> > **Limitations in formula expression.**：The reviewer places great emphasis on the accuracy of formulas. However, this paper repeatedly contains typos or parts that are not easy to follow. For example, the "isotropic scale" on Line 120 is evidently a scalar and should not be in bold. Similarly, in Eqn (2), the symbol p′p'p′ has not appeared earlier and is not explained in the surrounding text. In Eqn (3), it is clear that the outputs fradf\_{\\text{rad}}frad and fgeof\_{\\text{geo}}fgeo should be vectors. The introduction of Eqn (10) is overly casual; although the reviewer can infer the intended meaning, such writing issues should not appear in a NeurIPS paper.
>
> We sincerely appreciate the reviewer's careful reading and valuable feedback regarding the mathematical presentation. We will make the following revision of our manuscript to improve clarity:
>
> 1. Isotropic scale will be properly denoted as scalar $s$ (non-bold).
> 2. In Eqn (2),  $p'$ will be replaced with consistent notation $p_i$.
> 3. Vector outputs $f_{rad}$ and $f_{geo}$ will be bolded in Eqns (3) and (6).
> 4. Eqn (10) will be properly introduced with detailed explanation: “The final background color $C_{bg}$ combines the background point color $c_s$ with remaining visibility after accumulating the foreground Gaussians: $C_{bg}=\prod_{i=1}^N (1-\alpha_i) c_s, $ where $\alpha_i$ represents the opacity of the i-th Gaussian along the ray.”
> 5. We will conduct a thorough review of the entire manuscript to correct all typographical errors, including but not limited to the instance noted in Line 242.
>
> ---
>
> > **Activation function on scale**: It is not common to use a sigmoid function for the scale value in 3DGS. Could the authors explain why they choose a sigmoid function instead of the original exponential function used in 3DGS?
>
> We chose the sigmoid activation for scale values primarily for **numerical stability**. While the exponential function used in standard 3DGS is unbounded above and can grow rapidly, the sigmoid provides bounded outputs that are more suitable for neural field optimization. This is particularly important in our framework where scale predictions come from learned neural fields rather than being directly optimized. We observed experimentally that the exponential activation can lead to training instability, while sigmoid activation maintained stable optimization. We also note this design choice aligns with Scaffold-GS, which similarly employs sigmoid activation for scales.
>
> ---
>
> > **Report of the number of explicit Gaussians**: The reviewer believes that when discussing memory-efficient scene representation, it is important not only to report the final memory size but also to include the number of explicit Gaussians. This would allow readers to understand whether the proposed method reduces memory usage primarily by decreasing the data per Gaussian or also by reducing the number of points used for scene representation.
>
> We thank the reviewer for this valuable suggestion. The significant memory savings of HyRF come from both decreased per-Gaussian storage and reduced number of Gaussians. As stated in the paper, HyRF only stores 8 parameters per-Gaussian, in contrast to 59 parameters as in 3DGS. Moreover, HyRF naturally converges to fewer Gaussians while maintaining quality.
>
> As shown in the attached table, HyRF achieves a 24-45% reduction in the number of explicit Gaussians compared to 3DGS on three dataset (MipNeRF360, Tanks\&Temples and DeepBlending), without additional pruning techniques. We hypothesize this reduction of number of Gaussians stems from two key factors:
>
> 1. Faster convergence during training, reducing the need for aggressive densification.
> 2. The neural field's ability to represent view-dependent effects without requiring excessive Gaussians.
>
> | Dataset | MipNeRF360 | Tanks\&Temples | DeepBlending |
> | :---- | :----: | :----: | :----: |
> | 3DGS | 3.31M | 1.84M | 2.81M |
> | HyRF | 2.52M | 1.01M | 1.74M |
>
> These combined optimizations enable HyRF's significant memory savings. We will include this analysis in our revision to better demonstrate our method's advantages.
>
> ---
>
> > **Inefficiency on web platforms or low-end consumer GPUs:** The authors have made efforts in methodology and engineering to enable HyRF to achieve FPS comparable to vanilla 3DGS on an Nvidia 3090 GPU. However, since HyRF relies on Neural Fields during rendering, and Neural Fields tend to have low execution efficiency on web platforms or low-end GPUs, it is likely that HyRF's rendering efficiency on such devices will be significantly lower than that of vanilla 3DGS. This is a potential issue but does not detract from the fact that this is a well-rounded and meaningful research paper. This point is simply raised to highlight potential challenges in practical applications.
>
> We appreciate the reviewer's insightful observation regarding computational requirements. Indeed, the neural field components in HyRF currently benefit from high-end GPUs for high rendering speed. While recent advances like \[1\] have demonstrated efficient neural field rendering on mid-range GPUs (e.g., RTX 3060), achieving comparable efficiency on web platforms or integrated graphics remains an open challenge for the community. We hope future research in this direction can further bridge this efficiency gap, as it would significantly enhance the practical applicability of neural field based approaches.
>
> ---
>
> \[1\] City-on-Web: Real-Time Neural Rendering of Large-Scale Scenes on the Web. ECCV2024.

---

> ### Author Response · Authors · 2025-08-06
>
> Dear Reviewer,
>
> Thank you for your valuable insights and thoughtful suggestions on our work. We sincerely appreciate the time and effort you dedicated to reviewing our paper.
>
> Please let us know if our responses have adequately addressed your concerns or if there are any additional points you’d like us to consider. Your feedback is invaluable to us, and we are happy to make further revisions to improve our work.
>
> Thank you again for your constructive review.
>
> Best regards,
>
> Authors

---

> > ### Comment · Reviewer_Lw1w · 2025-08-07
> > **Response to Rebuttal**
> >
> > I appreciate the authors' thorough and careful rebuttal, which has addressed most of my concerns. Regarding the novelty issue, since more than one reviewer has recognized this paper's contributions, I've decided not to be overly strict on this point. However, I still want to highlight a limitation: since this work involves neural fields as a key component for scene representation, it cannot achieve efficient real-time rendering on web platforms or low-end consumer GPUs. I hope the authors will clearly state this limitation in the final version.
> >
> > Finally, I've decided to raise my rating from borderline reject to borderline accept.

---

> > > ### Author Response · Authors · 2025-08-07
> > >
> > > We sincerely appreciate the reviewer’s constructive feedback and the recognition of our efforts in addressing the concerns raised during the review process.
> > > We acknowledge the limitation on real-time rendering for web platforms or low-end GPUs and we will explicitly state this point in the limitation section in the revised version. Thank you again for your valuable insights, which have strengthened our manuscript.

---

### Official Review · Reviewer_7B5r · 2025-07-02

**Clarity:** 3
**Significance:** 2
**Originality:** 2
**Rating:** 4
**Confidence:** 3

**Summary:**

The paper combine explicit 3D Gaussians and neural fields for efficient neural rendering. The decomposition of geometry and appearance field helps to compress the parameters needed for view-dependent effects, compared to per Gaussian spherical harmonics parameters. The proposed method reduces the computational overhead by decomposing the foreground and background. It achieves 20X compression rate compared to original 3DGS and maintain real-time performance.

**Questions:**

1. The paper proposes to represent the anisotropic information with MLP and integration of neural field and explicit Gaussian, instead of high rank per Gaussian spherical harmonics parameters. However, ablation is missing: The memory footprint comparison between "MLP and integration of neural field and explicit Gaussian" and "high rank per Gaussian spherical harmonics parameters" as in original 3DGS.
2. Environment decomposition is common in street scenes, would be great to evaluate the proposed method in street scenes.

**Ethical Concerns:**

["NO or VERY MINOR ethics concerns only"]

**Final Justification:**

Thanks for addressing most of my concerns. I will change to positive rating.

**Limitations:**

See weakness above.

**Paper Formatting Concerns:**

Paper formatting is good.

**Quality:**

3

**Strengths And Weaknesses:**

Strength:

1. The results in supplementary shows a good novel-view synthesis comparison between baseline.

2. The decomposition of Geometry and Radiance/Appearance-related field is effective and is qualitatively verified in Figure 5.

3. The application of representing the view-dependent information with MLP instead of spherical harmonics for compression seems interesting to me.

Weakness:

1. The core contribution in line 68-70. The concept of geometry and rich appearance decomposition shares a similar idea with [1]. Would be great to have discussions.
2. The core contribution in line 70-71 is not entirely new in this domain, especially in GS for autonomous driving, for example [2,3,4] uses sky environment decomposition in their scene modeling. This might reduce the novelty from some extents.
3. The ablation of explicit gaussian in Line 259 is little confusing. It seems that the explicit  gaussian is essential to provide "position, explicit scale, explicit opacity, explicit color" in Figure2. How to inference the decomposed field if there is no explicit gaussian to provide "position"?
4. Related work on 3DGS compression is not compared, for example [5].

[1] https://arxiv.org/pdf/2003.09852
[2] https://arxiv.org/pdf/2311.18561
[3] https://arxiv.org/pdf/2401.01339
[4] https://arxiv.org/pdf/2405.18416
[5] https://arxiv.org/abs/2403.14530

---

> ### Author Rebuttal · Authors · 2025-07-31
>
> > The core contribution in line 68-70. The concept of geometry and rich appearance decomposition shares a similar idea with \[1\]. Would be great to have discussions.
>
> We would like to clarify the differences between \[1\] and our work in both motivation and model designs for the geometry-appearance decomposition:
>
> 1. **Different motivations.** \[1\] decouples geometry and appearance primarily for *efficiency*, as their geometry network is queried repeatedly using sphere tracing to get surface positions, before evaluating BRDF using the appearance network. In contrast, our decomposition aims to improve accuracy, as we empirically observed that a single network struggles to jointly learn Gaussian geometry and appearance attributes, likely due to their weak correlation (see L123-126 in our paper).
> 2. **Different model designs.** \[1\] models a signed distance function (SDF) for continuous surface modeling in the geometry network and a BRDF representation in the appearance network. In contrast, our method models per-Gaussian properties: opacity, scale, and rotation via the geometry network, and per-Gaussian colors via the appearance network.
>
> ---
>
> > The core contribution in line 70-71 is not entirely new in this domain, especially in GS for autonomous driving, for example \[2,3,4\] uses sky environment decomposition in their scene modeling. This might reduce the novelty from some extents.
>
> We appreciate the reviewer bringing up these relevant comparisons. Our background decomposition technique differs from prior sky modeling approaches in three key aspects:
>
> 1. **Memory Efficiency.** Unlike \[2,3\] that require storing cube maps (adding memory overhead), our appearance field predicts backgrounds without storage costs.
> 2. **No resolution limit.** While \[2,3\] are limited by cube map resolution, our neural field provides continuous background prediction.
> 3. **Improved Formulation.** In contrast to \[4\] which treats the background as points at infinity, our method instead constructs a background sphere and computes ray-sphere intersections as background points, which enhances rendering performance.
>
> The attached table shows a comparison of the three methods on Tanks & Temples dataset (*train* scene), where *Cube Maps* denotes the cube map approach in \[2, 3\], *Infinite Points* denotes the points-at-infinity approach in \[4\] and *Ours* denotes the background rendering method in our work. Our method achieves similar performance with the cube map approach \[2,3\] without introducing additional storage and offers higher visual quality compared with the points-at-infinity approach \[4\].
>
> |  | PSNR | SSIM | LPIPS | Size (MB) |
> | :---- | :----: | :----: | :----: | :----: |
> | Cube Maps | 22.01 | 0.803 | 0.213 | 48 |
> | Infinite Points | 21.87 | 0.792 | 0.221 | 37 |
> | Ours | **22.12** | **0.806** | **0.212** | 37 |
>
> ---
>
> > The ablation of explicit gaussian in Line 259 is little confusing. It seems that the explicit gaussian is essential to provide "position, explicit scale, explicit opacity, explicit color" in Figure2. How to inference the decomposed field if there is no explicit gaussian to provide "position"?
>
> To clarify, in our ablation experiment (Line 259), we maintain the explicit Gaussian positions as they are indeed necessary for spatial querying of the neural fields, while removing all other explicit properties (scale, opacity, and color) to evaluate the effectiveness of learning these properties through the neural fields only.
>
> We acknowledge this wasn't sufficiently clear in the original text and will revise Line 259 to explicitly state: "In Table 5, we evaluate the impact of removing all explicit Gaussian properties except positions, which are retained as they are required for neural field queries." We hope this revision can improve the manuscript's clarity.
>
> ---
>
> > Related work on 3DGS compression is not compared, for example \[5\].
>
> We thank the reviewer for pointing this out. We have actually included comparisons with \[5\] in Table 4 (denoted as Chen et al.) of our manuscript. We will make this comparison and discussion clearer in our revision.
>
> ---
>
> > The paper proposes to represent the anisotropic information with MLP and integration of neural field and explicit Gaussian, instead of high rank per Gaussian spherical harmonics parameters. However, ablation is missing: The memory footprint comparison between "MLP and integration of neural field and explicit Gaussian" and "high rank per Gaussian spherical harmonics parameters" as in original 3DGS.
>
> We thank the reviewer for this important suggestion. We provide an ablation study that compares two approaches (*SH Coefficients* for "high rank per Gaussian spherical harmonics parameters" and *Hybrid* for "MLP and integration of neural field and explicit Gaussian") for view-dependent appearance modeling:
>
> |  | PSNR | SSIM | LPIPS | Size (MB) |
> | :---- | :----: | :----: | :----: | :----: |
> | SH Coefficients | 30.12 | 0.908 | 0.243 | 267 |
> | Hybrid (Ours) | **30.37** | **0.910** | **0.241** | **34** |
>
> Our hybrid approach not only achieves significant reduction in model size, but also achieves slightly better visual quality compared with using SH coefficients. This comparison demonstrates that our hybrid approach provides a compact and more powerful way in modeling view-dependent appearance.
>
> ---
>
> > Environment decomposition is common in street scenes, would be great to evaluate the proposed method in street scenes.
>
> To evaluate HyRF's performance in street scenes, we conducted experiments on the KITTI \[6\] dataset (2011\_09\_26\_drive\_0002 sequence), with the following results:
>
> |  | PSNR | SSIM | LPIPS | Size (MB) |
> | :---- | :----: | :----: | :----: | :----: |
> | 3DGS | 19.37 | 0.665 | **0.272** | 472 |
> | HyRF (w/o background) | 19.42 | 0.660 | 0.273 | 36.7 |
> | HyRF (full) | **19.56** | **0.667** | 0.273 | **36.4** |
>
> Our method achieves similar visual quality compared with 3DGS while being over 10 times smaller in model size. After adding the background rendering technique, our complete method shows consistent quality improvements, particularly for distant objects and sky regions. This quantitative comparison, together with additional qualitative comparisons on the street scenes, will be added to our revised version.
>
> ---
>
> \[1\] Multiview Neural Surface Reconstruction by Disentangling Geometry and Appearance. NeurIPS 2020\.
> \[2\] Periodic Vibration Gaussian: Dynamic Urban Scene Reconstruction and Real-time Rendering. ArXiv.
> \[3\] Street Gaussians: Modeling Dynamic Urban Scenes with Gaussian Splatting. ECCV 2024\.
> \[4\] 3D StreetUnveiler with Semantic-aware 2DGS \-- a simple baseline. ICLR 2025\.
> \[5\] HAC: Hash-grid Assisted Context for 3D Gaussian Splatting Compression. ECCV 2024\.
> \[6\] Vision meets Robotics: The KITTI Dataset. IJRR 2013\.

---

> ### Author Response · Authors · 2025-08-06
>
> Dear Reviewer,
>
> Thank you for your valuable insights and thoughtful suggestions on our work. We sincerely appreciate the time and effort you dedicated to reviewing our paper.
>
> Please let us know if our responses have adequately addressed your concerns or if there are any additional points you’d like us to consider. Your feedback is invaluable to us, and we are happy to make further revisions to improve our work.
>
> Thank you again for your constructive review.
>
> Best regards,
>
> Authors

---

### Decision · Program_Chairs · 2025-09-17

**Decision:**

Accept (poster)

**Comment:**

This paper initially received mixed feedback, but after the rebuttal and discussion phase, the majority of reviewers shifted to a positive stance. The method was generally well-received for its clarity and potential impact, and the consensus among reviewers is to recommend acceptance.

One point that emerged during the AC–reviewer discussion concerns missing citations and positioning. In particular, the paper does not mention or discuss a critical related work, Loco-GS:

*“Locality-aware Gaussian Compression for Fast and High-quality Rendering,” Shin et al., ICLR 2025,*

which has been publicly available since early this year and presents conceptually related ideas. While the proposed HyRF approach includes differences, such as decoupling geometry and color via separate iNGPs, visibility pre-culling, and background rendering, there is also clear methodological overlap in how instant-NGP is combined with 3DGS. The authors should explicitly acknowledge this prior work and clarify the distinctions for the camera-ready.